# Lsd1 safeguards T-cell development via suppressing endogenous retroelements and interferon responses

Miaoran Xia[1],*, Bingbing Wang[1,2,3,4,5,*], Wujianan Sun[6,*], Dengyu Ji[1], Hang Zhou[1], Xuefeng Huang[1,2,3,4,5], Minghang Yu[1,2,3,4,5], Ziyang Su[1,2,3,4,5], Ping Chen[1], Kun Qu[6,7], Xi Wang[1,2,3,4,5]

The histone demethylase Lsd1 has been shown to play multiple essential roles in mammalian biology. However, its physiological functions in thymocyte development remain elusive. We observed that the specific deletion of Lsd1 in thymocytes caused significant thymic atrophy and reduced peripheral T cell populations with impaired proliferation capacity. Single-cell RNA sequencing combined with strand-specific total RNA-seq and ChIP-seq analysis revealed that ablation of Lsd1 led to the aberrant derepression of endogenous retroelements, which resulted in a viral mimicry state and activated the interferon pathway. Furthermore, the deletion of Lsd1 blocked the programmed sequential down-regulation of CD8 expression at the DP→CD4+CD8lo stage and induced an innate memory phenotype in both thymic and peripheral T cells. Single-cell TCR sequencing revealed the kinetics of TCR recombination in the mouse thymus. However, the preactivation state after Lsd1 deletion neither disturbed the timeline of TCR rearrangement nor reshaped the TCR repertoire of SP cells. Overall, our study provides new insight into the function of Lsd1 as an important maintainer of endogenous retroelement homeostasis in early T-cell development.

## Introduction

T-cell development in the thymus is a precise and orderly regulated multistep process. Briefly, CD4−CD8− double-negative (DN) thymocytes can be further subdivided into sequential stages, including the DN1 (CD44+CD25−), DN2 (CD44+CD25+), DN3 (CD44−CD25+), and DN4 (CD44−CD25−) stages. The cells then differentiate into the CD8+TCRlo immature single-positive (ISP) stage and the CD4+CD8+ double-positive (DP) stage and finally develop into mature CD4+ or CD8+ single-positive (SP) T cells that migrate to the periphery ([1]). The chromatin state (the packaging of DNA with histone proteins) and its epigenetic regulation modulators play critical roles in the development of thymocytes. Histone posttranslational modifications include phosphorylation, acetylation, ubiquitinylation, methylation, and others ([2]). It is critically involved in almost all aspects of T-cell biology, including lineage commitment, development, activation, differentiation, and memory formation ([3]). For example, the loss of histone methyltransferase *Ezh2* has been found to impede T-cell differentiation in the DN phase ([4], [5]) but not change the production of mature peripheral T cells ([6]). However, Vasanthakumar et al reported that conditional KO of *Ezh2* and other polycomb repressive complex 2 (PRC2) components (*Suz12* and *Eed*) at the DP stage did not alter the subsequent development of αβ or γδ T-cell development in the thymus and spleen ([7]). Tamoxifen-induced deletion of the histone deubiquitinase *Bap1* in adult mice resulted in severe thymic atrophy with a block at the DN3 stage and complete loss of the T-cell lineage ([8]). It has also been reported that the PRC1, PRC2, histone methyltransferase G9a, and a variety of lncRNAs influence the differentiation and maintenance of T helper cells by epigenetically regulating transcriptional programs associated with different T-cell subsets ([9]). Understanding the mechanisms of epigenetic regulation of T-cell development will have important implications for T-cell biology and translational therapy.

Histone methylation has been believed for a long time to be irreversible until Lsd1 (Lysine-specific demethylase 1A, encoded by *Kdm1a* in mice) was identified to be a bonafide histone demethylase in 2004 ([10]). The enzyme can demethylate histone H3 on Lys4 as a transcription corepressor or on Lys9 as a transcription coactivator. It is essential for a wide range of biological events. Deletion of the gene results in developmental arrest and the death of mouse embryos ([11], [12], [13]). It is widely reported to be an oncogene that promotes cancer cell proliferation, migration, and invasion ([14], [15]), and has been found to repress antitumor T-cell immunity ([16]).

[1]Department of Immunology, School of Basic Medical Sciences, Beijing Key Laboratory for Tumor Invasion and Metastasis, Capital Medical University, Beijing, China [2]Institute of Infectious Diseases, Beijing Key Laboratory of Emerging Infectious Diseases, Beijing Ditan Hospital, Capital Medical University, Beijing, China [3]Beijing Institute of Infectious Diseases, Beijing, China [4]National Center for Infectious Diseases, Beijing Ditan Hospital, Capital Medical University, Beijing, China [5]Department of Oncology, Capital Medical University, Beijing, China [6]Department of Oncology, The First Affiliated Hospital of USTC, School of Basic Medical Sciences, Division of Life Sciences and Medicine, University of Science and Technology of China, Hefei, China [7]Institute of Artificial Intelligence, Hefei Comprehensive National Science Center, Hefei, China

Correspondence: xiwang@ccmu.edu.cn; qukun@ustc.edu.cn; chenping@ccmu.edu.cn
*Miaoran Xia, Bingbing Wang, and Wujianan Sun contributed equally to this work

Kerenyi et al reported that inducible deletion of Lsd1 in early hematopoietic stem cells not only compromised early hematopoietic differentiation but also strongly interfered with terminal granulocytic and erythroid differentiation (17). Subsequently, several groups have reported that Lsd1 is required for the germinal center formation and the humoral immune response (18, 19, 20). Recently, Lsd1 was found to regulate multiple repressive gene programs during T-cell development (21). The study showed that Lsd1 represses genes that are normally down-regulated during the DN-to-DP transition, such as stem cell-related genes and checkpoint molecule genes. However, only a small fraction of H3K4me3 sites, rather than H3K4me1/2, were found to increase, which was not sufficient for the up-regulation of the observed genes, as concluded by the authors. The physiological role of Lsd1 in early T-cell development remains unclear.

In this study, we found that conditional deletion of Lsd1 at the DN stage led to thymic atrophy and decreased peripheral T-cell populations with impaired proliferation capacity. We further verified that the "IFN response" pathway was dramatically up-regulated throughout all developmental stages by single-cell RNA sequencing (scRNA-seq) analysis. Moreover, we identified endogenous retroelements (EREs) as direct targets of Lsd1, whose H3K4me1 and H3K4me2 modifications were increased in KO mice rather than IFN-stimulated genes (ISGs). Notably, after Lsd1 depletion, CD8 expression failed to be down-regulated in time at the DP→CD4$^+$CD8$^{lo}$ stage, which could have an impact on T-cell function. Single-cell TCR sequencing revealed the kinetics of TCR recombination in the mouse thymus. However, the preactivation state after Lsd1 deletion neither disturbed the timeline of TCR rearrangement nor reshaped the TCR repertoire of SP cells. Together, the results indicate that Lsd1 functions to prevent aberrant stimulation of the IFN pathway by repressing endogenous EREs and plays a critical role in maintaining normal T-cell development.

## Results

### Ablation of Lsd1 disrupts the proliferation capacity of T cells

To elucidate the function of Lsd1 in thymocyte development, we crossed Lsd1$^{fl/fl}$ mice with Lck-Cre mice (Fig S1A), which drives Cre recombinase expression via the proximal lymphocyte-specific protein tyrosine kinase (Lck) promoter and effectively deletes the floxed gene fragment at the DN stage (Fig S1B). To exclude the possible effect of Lck-driven Cre expression on T-cell development as reported (22), we used heterozygotes (Lsd1$^{wt/fl}$Lck-Cre) as littermate controls (referred to hereafter as controls). The expression of Lsd1 in thymocytes was greatly lower in Lsd1$^{fl/fl}$Lck-Cre mice (referred to hereafter as KO) than in the controls, showing the high efficiency of Lsd1 deletion (Fig S1C). By performing Western blots on nuclear extracts, we observed increased H3K4me2 modification in KO thymocytes, whereas the H3K4me1/3, H3K9me1/2/3, and H3K27ac modifications were not changed (Fig S1C). This was further confirmed by flow cytometry. The results showed that H3K4me2 was increased explicitly in Lsd1-deleted T lineage cells beginning in the DN stage, whereas the H3K4me1 level was not changed (Fig S1D).

We observed that at 6~10 wk of age, KO mice had a markedly smaller thymus (Fig 1A) with a 35% reduction in weight (Fig 1B), a 50% reduction in absolute cell numbers (Fig 1C), and an abnormal architecture with remarkably atrophic cortex and expanded medullary regions (Fig 1D). Increased cell apoptosis in the Lsd1-KO thymus had also been detected (Fig S1E). Flow cytometry analysis showed that, compared with control mice, KO mice exhibited no changes in the percentages of CD4$^-$CD8$^-$ cells and CD4$^+$CD8$^+$ cells but exhibited a considerable decrease in the percentages of CD4$^+$CD8$^-$ cells and a relative increase in the percentages of CD4$^-$CD8$^+$ cells in thymocytes (Fig 1E–G).

We further investigated whether the loss of Lsd1 affects migration of mature T cells to the periphery. As shown in Fig S2, the total cell numbers of spleens and lymph nodes were not affected (Fig S2A). However, the frequencies of CD3$^+$ T cells in the spleens (Fig 2A) and lymph nodes (Fig S2B) from KO mice were significantly decreased compared with those of control mice. Moreover, we found that the expression levels of CD3 on peripheral T cells were down-regulated (Fig 2B). The percentages of both CD4$^+$ and CD8$^+$ T cells were greatly reduced in the spleen (Fig 2C) and lymph nodes (Fig S2C). This was consistent with a previous report in which Lsd1 was deleted in immature DN thymocytes using Cd2-Cre mice (21). The authors attributed this finding to the reduced S1pr1 required for thymocyte emigration on mature SP cells. However, it was interesting that although the CD4/CD8 ratio was altered inside the thymus in KO mice (Fig 1G), it was not affected in spleens (Fig 2D) and lymph nodes (Fig S2D). Notably, the proliferation capacities of both peripheral CD4$^+$ and CD8$^+$ T cells under anti-CD3 and anti-CD28 stimulation were impaired after Lsd1 deletion (Fig 2E), which may have also contributed to the significant decrease in the number of peripheral T cells. In summary, the loss of Lsd1 disrupted the numbers and proliferation of mature T cells in the periphery.

More interestingly, we observed that the specific deletion of Lsd1 in thymocytes at the DN stage also affected the development of other immune cells in the thymus. The percentages and absolute numbers of thymic B (CD4$^-$CD8$^-$B220$^+$) cells and thymic NK cells (CD4$^-$CD8$^-$CD122$^+$NK1.1$^+$) were increased significantly in KO mice compared with control mice (Fig S3A and B). We examined whether the accumulation of B or NK cells resulted from transdifferentiation of T-precursors or reactive hyperplasia. YFP was used as an indicator of the activation of Lck-Cre recombinase, which is believed to be specifically expressed in T-lineage cells. If a cell is derived from the T-cell precursor, it will express YFP and emit fluorescence. Our results showed that no YFP$^+$ B cells were observed in the thymus, bone marrow, spleen, and lymph nodes from KO mice, which indicated that no transdifferentiation from T to B cells occurred after Lsd1 deletion (Fig S3C). Of note, it was unexpected that thymic NK cells could also activate the Lck-Cre recombinase, as there were YFP$^+$ NK cells in the control thymus (Fig S3C). Considering the similar frequencies of YFP$^+$ thymic NK cells in KO mice compared with control mice, we speculated that it resulted from reactive hyperplasia but not transdifferentiation. Consistently, a previous investigation also observed increased numbers of B cells and NK cells in the thymus in Lsd1$^{fl/fl}$Cd2-Cre mice but not in Lsd1$^{fl/fl}$Cd2-Cre DN thymocyte cultures on OP9-DL1 cells (21). In summary, the observed increased numbers of other lineage cells were not caused by transdifferentiation from T cells. This may be caused by the change

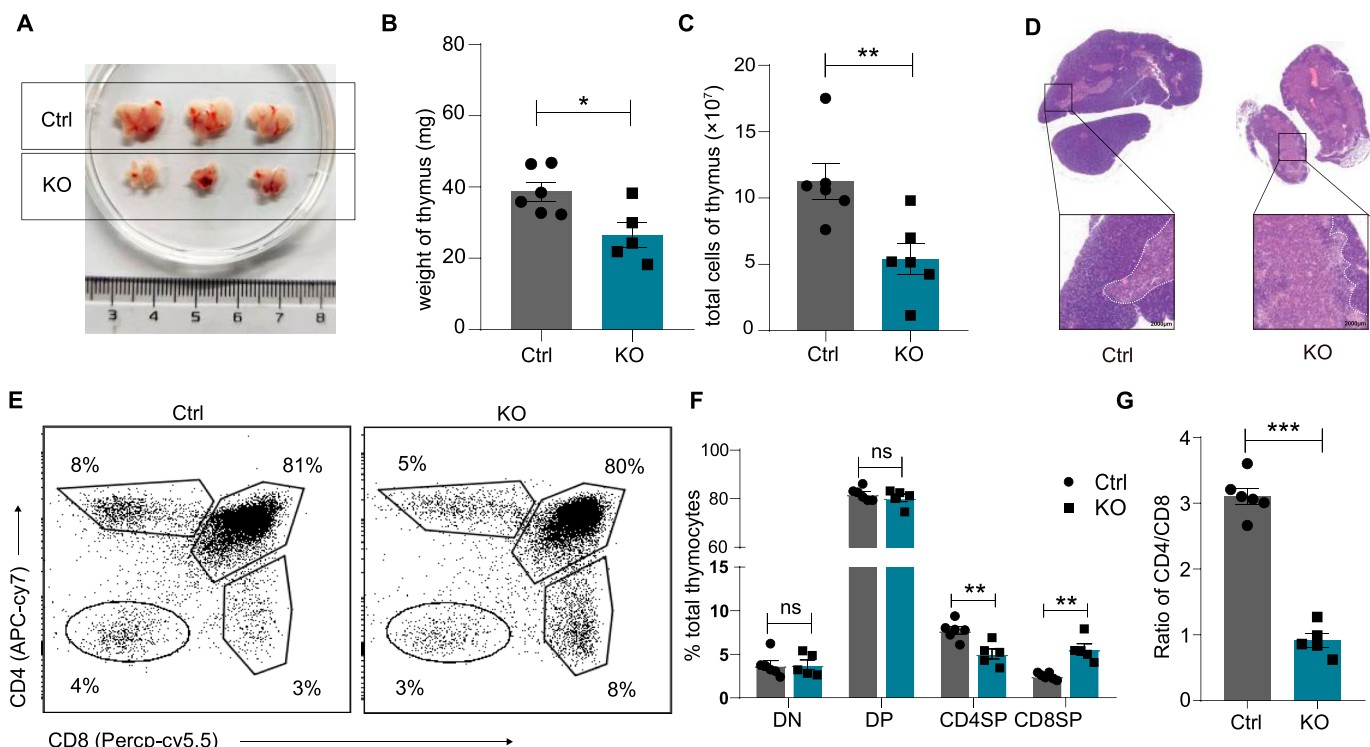

**Figure 1. Ablation of Lsd1 causes thymic atrophy.**
**(A)** Images of thymi from control (n = 3) and KO (n = 3) mice. **(B)** Weight of the thymi from control (n = 6) and KO (n = 5) mice. **(C)** Total cell numbers of thymocytes from control (n = 6) and KO (n = 6) mice. **(D)** Representative hematoxylin and eosin (H&E) staining of thymi from control and KO mice. Original magnification: ×4 (up), ×10 (down). Scale bars: 2,000 μm. **(E)** Representative CD4 versus CD8 staining of total live thymocytes from control and KO mice. **(F)** Percentages of the indicated populations in total thymocytes from control (n = 6) and KO (n = 5) mice. DN: CD4⁻CD8⁻ double-negative cells, DP: CD4⁺CD8⁺ double-positive cells, CD4SP: CD4⁺CD8⁻ single-positive cells, CD8SP: CD4⁻CD8⁺ single-positive cells. **(G)** Ratios of CD4SP versus CD8SP subsets in the thymi from control (n = 6) and KO (n = 5) mice. Control: *Lsd1*^wt/fl^*Lck-Cre*, KO: *Lsd1*^fl/fl^*Lck-Cre*. Cumulative data are means ± SEMs. *$P < 0.05$, **$P < 0.01$, ***$P < 0.001$, ns, no significance, as determined by unpaired *t* test.

in the thymic microenvironment after the specific deletion of Lsd1 in thymocytes.

### Single-cell sequencing reveals that ablation of Lsd1 disturbs the programmed down-regulation of CD8 expression at the DP→CD4⁺CD8^lo stage

To better understand the effect of Lsd1 on T-cell development in the thymus, we profiled thymocytes from three control mice and three KO mice with scRNA-seq based on a 10× Genomics platform and obtained a total of 39,857 cells (Fig 3A). After filtering low-quality cells with abnormal numbers of expressed genes, high mitochondrial gene expression, and doublets, the remaining bioinformatically identified cells from all six thymi (38,966 cells), with an average of 1,583 genes per cell, were combined for downstream analysis.

We identified 28 clusters with distinct transcriptomic signatures using unbiased clustering and Uniform Manifold Approximation and Projection (UMAP) analysis. We identified the clusters of 11 cell types at different developmental stages of T thymocytes using SingleR with the ImmGen reference dataset (23). The cell types were ETP-DN3a, DN3b-ISP, T.DPbl (blasts), T.DPsm (small resting), T.DP69⁺ (early positive selection), T.SPinter (intermediate), T.CD4Th, T.CD8, T.CD4Treg, γδT and natural killer T (NKT) cells; other lineages,

namely, B cells and dendritic cells (DCs), were also observed (Fig 3B and C).

Accordingly, we observed similar subpopulation distributions in the thymi from control and KO mice (Fig 3D). Furthermore, we calculated the proportions of the different subgroups. Consistent with our flow cytometry analysis, we observed increased frequencies of other lineages (e.g., B cells and DCs) in the scRNA-seq data (Fig 3E). Considering the T-cell lineage, the proportions of Tregs and NKTs were increased, which was confirmed by flow cytometry (Fig S3D). However, the proportions of other T-cell subgroups showed no significant differences (Fig 3E). This result was striking because significant changes were observed in the percentages of CD4⁺ cells and CD8⁺ cells in flow cytometry analysis (Fig 1F). We then tried to determine what made the difference. Interestingly, we observed greatly increased CD8 expression and decreased CD4 expression in the T.SPinter cells (representing the intermediate CD4⁺CD8^lo stage) and CD4Th cells in the Lsd1-deleted mice (Fig 4A). CD4⁺CD8⁺ DP thymocytes differentiate into CD4⁺CD8^lo cells and then make a lineage choice to become either CD4⁺ or CD8⁺ SP T cells (Fig 4B) (24). The increased CD8 expression and decreased CD4 expression in the T.SPinter cells could have led them to be counted in the CD8⁺ SP gate in flow cytometry analysis (Fig 4C). However, in the scRNA-seq map, they were still defined as T.SPinter cells based on their general transcriptome. Consistently, the

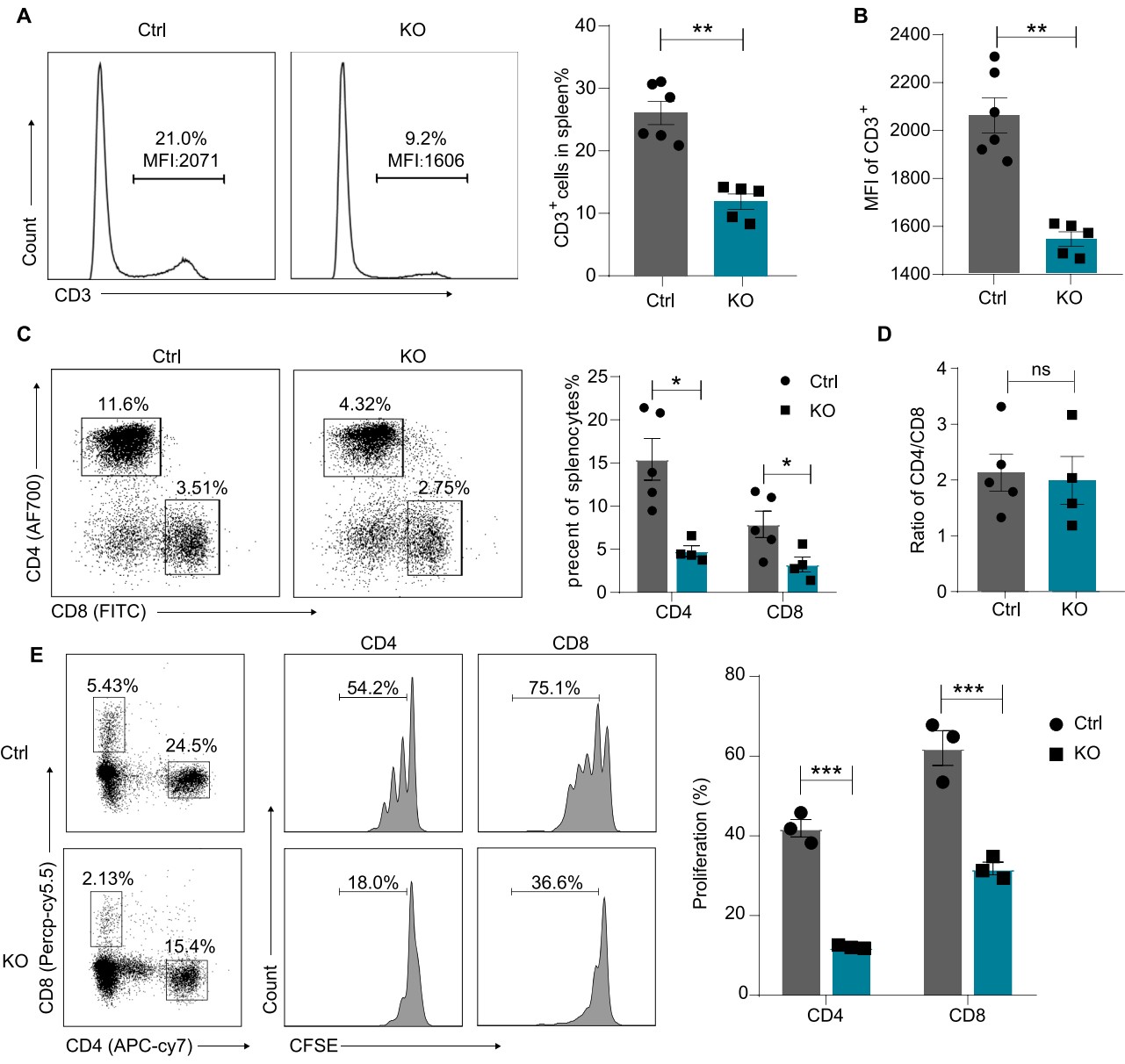

**Figure 2.  Decreased peripheral T cells with impaired proliferation capacity in Lsd1-deleted mice.**
**(A)** Representative histograms show the percentages of CD3⁺ cells in the spleens of control and KO mice. Frequencies of CD3⁺ T cells in the thymi from control (n = 6) and KO (n = 5) mice are statistically shown on the right. **(B)** The mean fluorescence intensity (MFI) of CD3 expression on thymocytes from control (n = 6) and KO (n = 5) mice. **(C)** Representative CD4 versus CD8 staining of splenocytes from control and KO mice. The cumulative data on the frequencies of CD4⁺ or CD8⁺ cells in the spleen from control (n = 5) and KO (n = 4) mice are shown on the right. **(D)** Ratio of CD4⁺ versus CD8⁺ subsets in the spleen. **(E)** Analysis of the proliferation capacities of splenic T cells. Splenocytes from control (n = 3) and KO (n = 3) mice were labeled with CFSE and stimulated with anti-CD3 and anti-CD28 in vitro. After 3 d of culture, the proliferation of CD4⁺ T cells and CD8⁺ T cells was examined by flow cytometry for the dilution of CFSE. Control: *Lsd1^{wt/fl}Lck-Cre*, KO: *Lsd1^{fl/fl}Lck-Cre*. Cumulative data are means ± SEMs. *$P$ < 0.05, **$P$ < 0.01, ***$P$ < 0.001, as determined by unpaired $t$ test.

proportions of CD4⁺/CD8⁺ cells in flow cytometry could be mimicked by adding T.SPinter cells to the CD8⁺ group for abundance analysis in the scRNA-seq data (Fig 4D). Furthermore, CCR9 was identified as a surface marker distinguishing T.SPinter cells from CD4Th cells (Fig 4E), consistent with that in the human thymus (25). To confirm this hypothesis, we stained thymocytes with anti-CCR9 antibodies. As expected, CCR9⁺ and CCR9⁻ subgroups were observed under the CD4⁺CD8⁻ gate by flow cytometry (Fig 4F). The CD4⁺CCR9⁺ subgroup

size was significantly decreased in KO mice, whereas the number of CD8⁺CCR9⁺ cells was increased.

Furthermore, we investigated the effect of CD8 up-regulation on T-cell development. It has been reported that constitutively expressed CD8 can promote most of the MHC-I-restricted thymocytes to develop into innate memory-like CD8⁺ T cells rather than redirecting them to the CD4 helper T-cell lineage (26). Consistently, we observed increased CD8⁺ T cells with an innate-memory

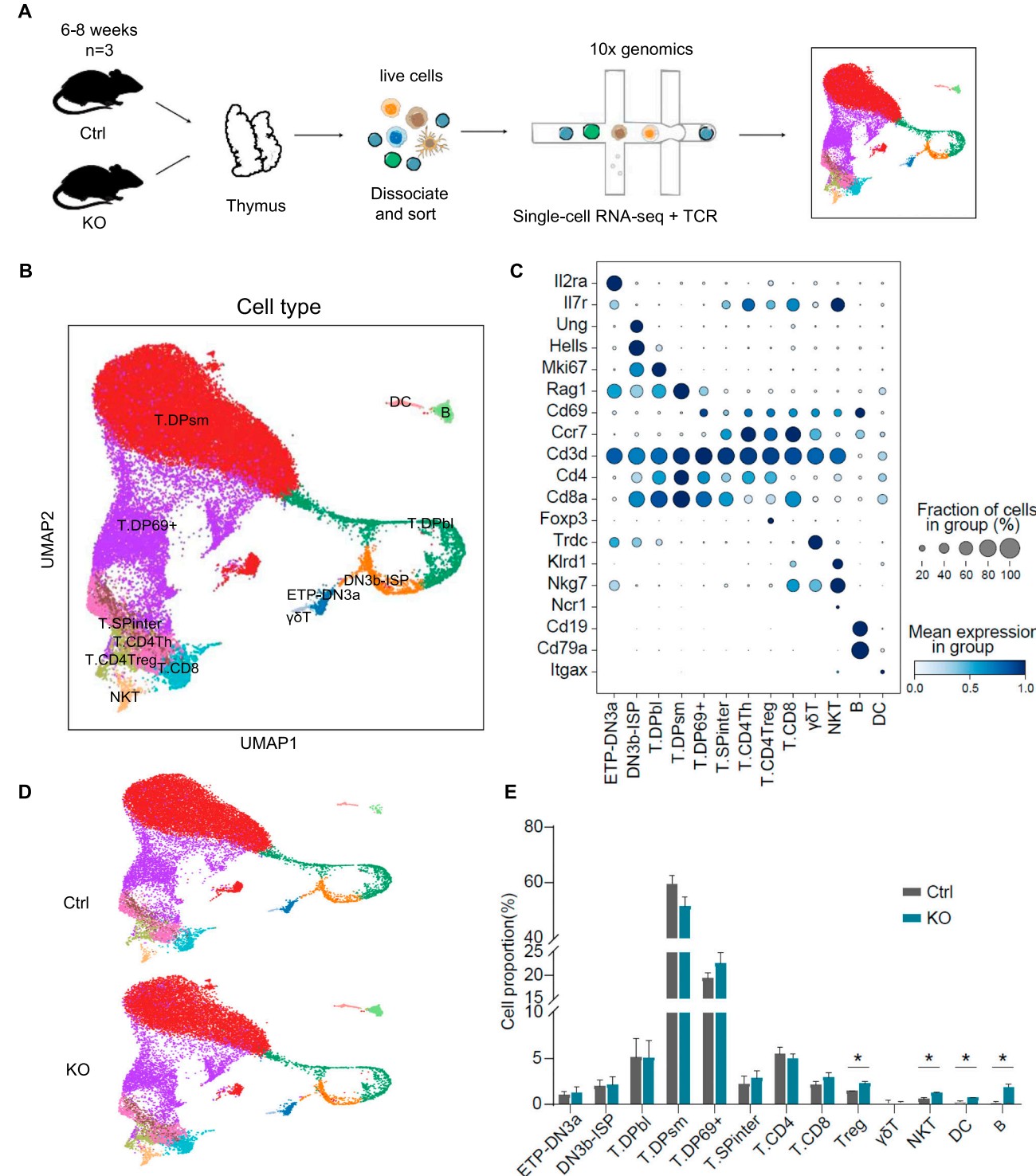

**Figure 3. scRNA-seq of thymocytes.**

**(A)** Schematic of procedures for the sorting and scRNA-seq of thymocytes. **(B)** UMAP visualization of scRNA-seq data for single cells derived from control (n = 3) and KO (n = 3) mice. Cells were color coded by cell type; each dot represents one cell. ETP, early T-cell progenitor; DN, double-negative cell; ISP, immature single-positive cell; T.DPbl, blasting double-positive T cell; T.DPsm, small resting double-positive T cell; T.DP69+, CD69+ double-positive T cell, early positive selection; T.SPinter, intermediate single positive T cell; T.CD4Th, CD4+ helper T cell; T.CD8, CD8+ T cell; T.CD4Treg, CD4+ regulatory T cell; NKT, natural killer T cell; DC, dendritic cell. **(C)** Expression patterns of marker genes for each cell subtype. The fraction of cells that expressed the marker genes is indicated by the size of the circle, as shown in the scale on the right. The means of the expression levels of marker genes are indicated by the color. **(D)** UMAP visualization of scRNA-seq data derived from control (up) and KO (down) mice. The cells are color coded by cell type; each dot represents one cell. **(E)** Cell proportions are shown by cell type in control (n = 3) and KO (n = 3) mice. Control: *Lsd1$^{wt/fl}$Lck-Cre*, KO: *Lsd1$^{fl/fl}$Lck-Cre*. Cumulative data are means ± SEMs. *$P < 0.05$, as determined by unpaired *t* test.

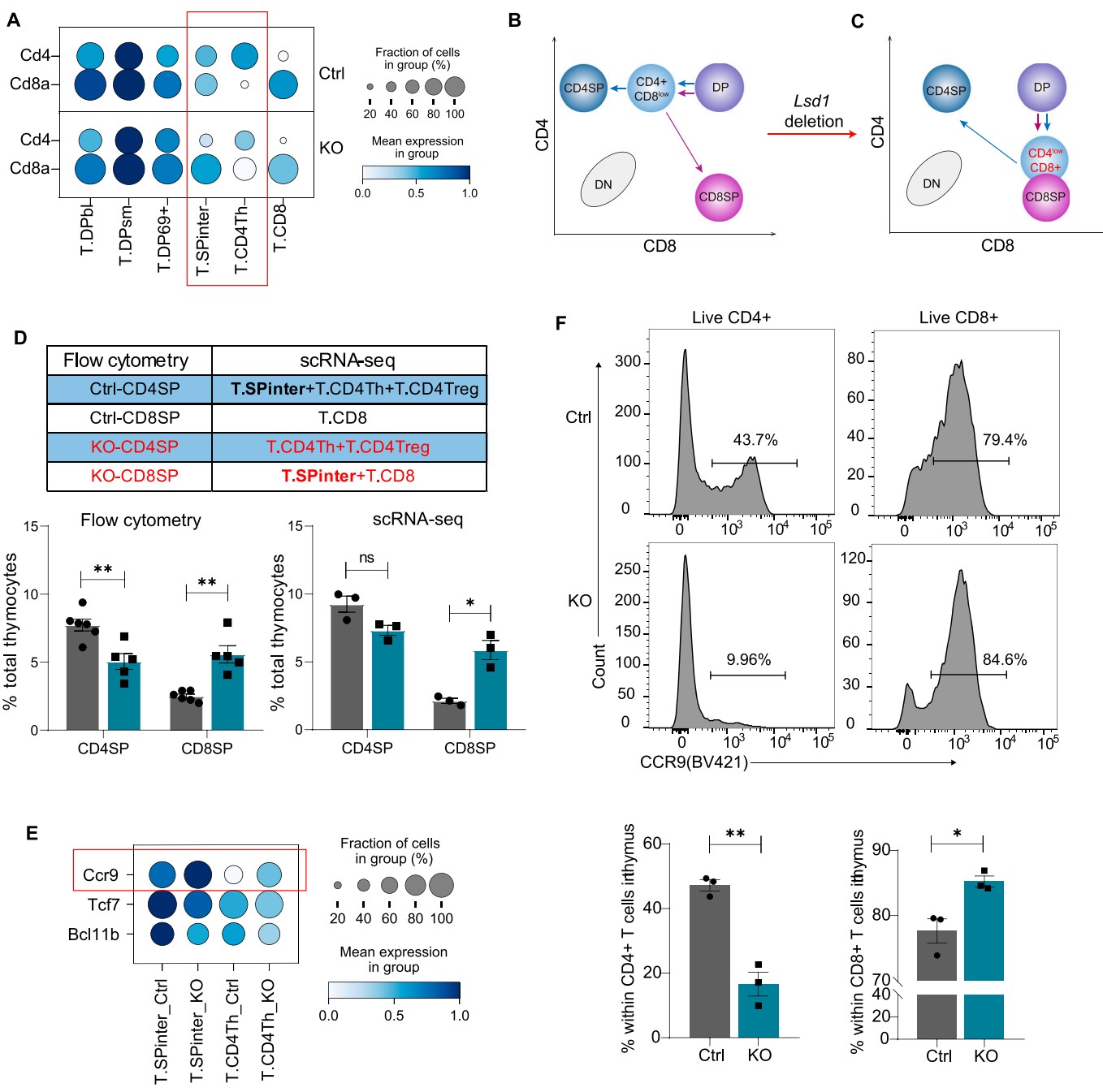

**Figure 4. Ablation of Lsd1 disturbs the programmed down-regulation of CD8 expression at the DP→CD4⁺CD8$^{lo}$ stage.**
**(A)** CD4 and CD8a expression in T-cell subgroups. **(B, C)** A model for the differentiation from DP to SP cells before (B) and after (C) the deletion of Lsd1. **(D)** CD4SP and CD8SP abundance in scRNA-seq (right) mimicking that in flow cytometry (left) by adding SPinter cells to the CD8⁺ group. **(E)** DEGs identified as marker genes distinguishing the T.SPinter and T.CD4Th cells. **(F)** CCR9⁺ T.SPinter cells in CD4⁺ (left) and CD8⁺ (right) thymocytes identified by flow cytometry. Statistical data are shown below. Control: $Lsd1^{wt/fl}Lck$-$Cre$, KO: $Lsd1^{fl/fl}Lck$-$Cre$. Cumulative data are means ± SEMs. *$P < 0.05$, **$P < 0.01$, as determined by unpaired $t$ test.

phenotype expressing CD44 and CXCR3 in KO thymocytes (Fig S4A), and significantly elevated innate-memory scores of the T.CD8 subgroup in scRNA-seq data (Fig S4B). Similarly, in a previous study on $Lsd1^{fl/fl}Cd2$-$Cre$ mice, the expression of innate memory T-cell–associated genes was increased in CD69⁻ thymocytes, CD8 SP thymocytes, and peripheral CD8⁺ T cells (21). The authors concluded that this resulted from the influence of Lsd1 on Bcl11b-mediated

genes because the phenotype was similar to that observed in Bcl11b-deficient mice, and most of the Bcl11b-repressed genes were up-regulated in $Lsd1^{fl/fl}Cd2$-$Cre$ CD69⁻ thymocytes (21). However, the interaction of Lsd1 and Bcl11b could not explain the change in CD8 expression. It appears that the innate memory phenotype may be generated through multiple mechanisms. In summary, the loss of Lsd1 disturbs the programmed down-regulation of CD8 expression

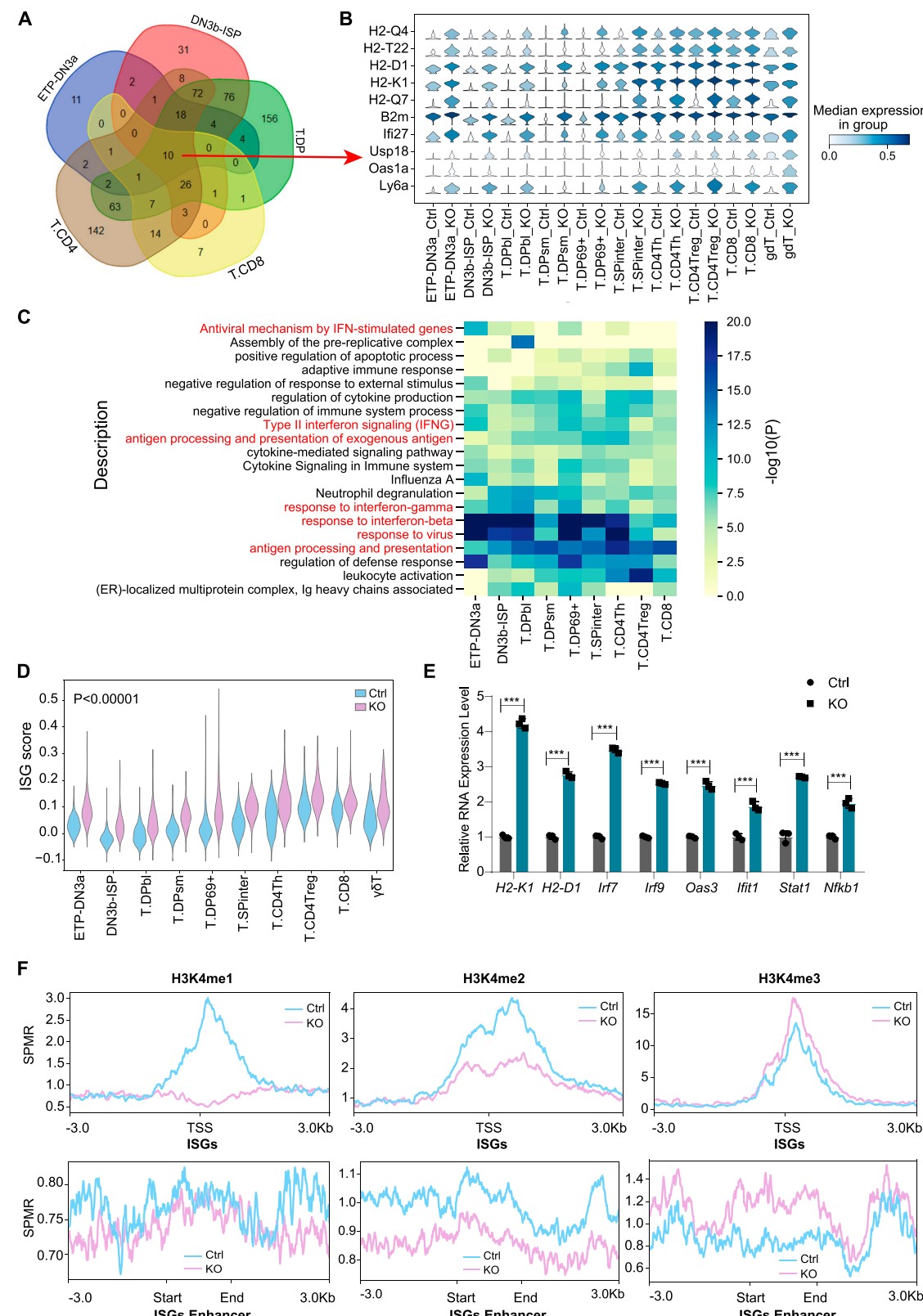

**Figure 5. Deletion of Lsd1 activates the IFN response in thymocytes.**
**(A)** Venn diagrams of the up-regulated genes after Lsd1 deletion in different T-cell subgroups. **(B)** Violin plots showing the expression of the 10 shared up-regulated genes in different T subgroups from control and KO mice. **(C)** Heatmap of the significantly enriched GO terms for the genes overexpressed in different T-cell subgroups from control and KO mice. **(D)** Violin plots show the ISG scores of each cell across different T-cell subgroups from control and KO mice. $P < 0.00001$ in all compared groups.

at the DP→CD4+CD8lo stage and promotes an innate memory phenotype in CD8+ T cells.

## Lsd1 regulates activation of the IFN signaling in thymocytes

To determine how Lsd1 regulates thymic T-cell development, we performed differentially expressed gene (DEG) and pathway enrichment analyses of the scRNA-seq transcriptome data. As shown in Fig 5A and B, we found 10 up-regulated genes (*H2-Q4*, *H2-T22*, *H2-D1*, *H2-K1*, *H2-Q7*, *B2m*, *Ifi27*, *Usp18*, *Osa1a*, and *ly6a*) shared by T-cell subgroups at different stages (ETP-DN3a, DN3b-ISP, T.DP, T.CD8, T.CD4). The number of down-regulated genes was much lower, and almost no shared genes were found. Surprisingly, among the shared up-regulated genes, six genes belonged to the MHC I complex, and the others were ISGs. Consistently, we found high enrichment of IFN-response pathways and antigen-processing pathways across all the T-cell subgroups (Fig 5C). Because the IFN pathway is a positive regulator of the MHC antigen-processing machinery (27), we can conclude that the DEGs and their related pathways are associated with "the activation of IFN response." We further evaluated the IFN response using an ISG module score (28). Interestingly, in the control thymocytes, the ISG score was high at the early ETP-DN3a stage. Then, it was down-regulated at the DN3b stage and gradually increased during the subsequent developmental stages. After the loss of Lsd1, ISG scores were significantly elevated in all T-cell stages (Fig 5D), indicating excessive pre-activation of the IFN pathway during the whole developmental process. Quantitative polymerase chain reaction (qPCR) analysis further confirmed the overexpression of MHC I molecules (*H2-k1*, *H2-d1*) and ISGs (*Irf7*, *Irf9*, *Oas3*, *Ifit1*, *Stat1*, *Nfkb1*) (Fig 5E). As known, T cells mainly secret IFNγ under stimulus (29). Increased expression of IFN-II (IFN-γ) was determined at both the mRNA and protein levels in KO mice compared with control mice (Fig S5A and B), whereas the IFN-I gene (*Ifna1*, *Ifnb*) and IFN-III gene (*Il28b*) were expressed at much lower levels and even showed some decrease in expression (Fig S5A) in KO thymocytes. In addition, there is a possibility that the abnormal up-regulation of IFNγ in the thymic environment also activated the related immune cells and caused increased numbers of other-lineage cells (B cells, NK cells, DCs) in the Lsd1-deleted thymus, as we observed in flow cytometry and scRNA-seq analysis. These data indicated that Lsd1 ablation was causing aberrant activation of IFNγ signaling. This was consistent with a previous report demonstrating that knockdown of Lsd1 in immature T cells results in the overexpression of genes involved in IFN/viral response-related functions, according to bulk RNA-seq of CD69− cells and CD69+ DP cells (21). Our data suggested that IFNγ signaling was overactivated in all stages of T cells.

We then investigated whether Lsd1 directly targets IFN-responsive genes to repress their expression. Lsd1 is known as an H3K4me1/2 demethylase that acts as a transcription corepressor. We evaluated whether Lsd1 directly regulates the modification of H3K4me1/2 at

IFN-responsive genes by analyzing publicly available chromatin immunoprecipitation sequencing (ChIP-seq) data obtained from thymocytes in which Lsd1 was knocked out at the DN stage by CD2-Cre recombinase (21). They found few changed H3K4me1/2 marks, and only a small fraction of H3K4me3 marks were increased. To be more specialized, we analyzed the H3K4me1/2/3 modifications of ISGs. As shown in Fig 5F, the H3K4me1 and H3K4me2 modifications of ISGs and the H3K4me2 marks of their enhancers were decreased after Lsd1 depletion. They exhibited only enhanced decoration with H3K4me3 marks at their enhancers. This result indicated that the up-regulation of ISGs could result from other epigenetic modifications. We conclude that the ISGs are not directly regulated by the H3K4me1/2 demethylase activity of Lsd1. In summary, the loss of Lsd1 activates the IFNγ response in thymocytes, whereas the overexpression of ISGs is not directly regulated by Lsd1.

## Deletion of Lsd1 derepresses EREs

We then attempted to determine what triggers the IFN response in Lsd1-deleted thymocytes. Other than the IFNγ response genes, the pathway enrichment analysis of scRNA-seq data showed up-regulation of the "response to the virus" pathway after Lsd1 loss (Fig 5C), suggesting that the activation of an upstream event, such as an RNA-sensing pathway, may play an important role. EREs account for ~40% of mammalian genomes, and the silencing of EREs is controlled by the state of histone methylation (30). We wondered if there was an abnormal transcription of EREs triggering IFN signaling after Lsd1 deletion. As our scRNA-seq focused on poly-A eukaryotic mRNAs, we further performed strand-specific total RNA-seq of thymocytes to detect noncoding RNAs. Increased numbers of transcripts in both sense and antisense directions from all ERE subfamilies were detected in the Lsd1-deleted thymocytes, including LTR-containing endogenous retroviruses (ERV1s, ERVKs, and ERVLs) and non-LTR elements (LINEs and SINEs) (Fig 6A and B). ChIP-seq analysis showed that the loci of up-regulated EREs had a great increase in H3K4me1 and H3K4me2 levels, and a mild increase in H3K4me3 levels in KO mice, which indicated that Lsd1 directly targets these EREs to regulate their expression (Fig 6C). The overexpression of EREs can contribute to the generation of dsRNAs, which trigger IFN signaling activation. In addition, we detected the expression levels of the dsRNA sensors Tlr3, Mda5 (encoded by *Ifih1*), and Rig-I (encoded by *Ddx58*) and the DNA sensors Sting (encoded by *Sting1*) and Cgas in Lsd1-deleted thymocytes and found that they were all increased as expected (Fig 6D). These data suggested that ablation of Lsd1 derepresses the transcription of a group of EREs, resulting in aberrant activation of the IFNγ response in mouse thymocytes. Consistently, Lsd1 has been reported to be an ERV suppressor in embryonic stem (ES) cells (30) and regulates the ERV-IFN pathway in melanoma cells (16).

---

**(E)** The expression of MHC I molecules (*H2-k1*, *H2-d1*) and ISGs (*Irf7*, *Irf9*, *Oas3*, *Ifit1*, *Stat1*, *Nfkb1*) in control and KO thymocytes analysed by qPCR. The error bars represent the SD between triplicates in one of three experiments. **(F)** H3K4me1, H3K4me2, and H3K4me3 ChIP-seq signals at ISG loci (up) and their enhancer regions (down) in control and KO thymocytes. Control: *Lsd1wt/fl Lck-Cre*, KO: *Lsd1fl/fl Lck-Cre*. Cumulative data are means ± SEMs. *P < 0.05, **P < 0.01, ***P < 0.001, ns, no significance, as determined by unpaired *t* test.

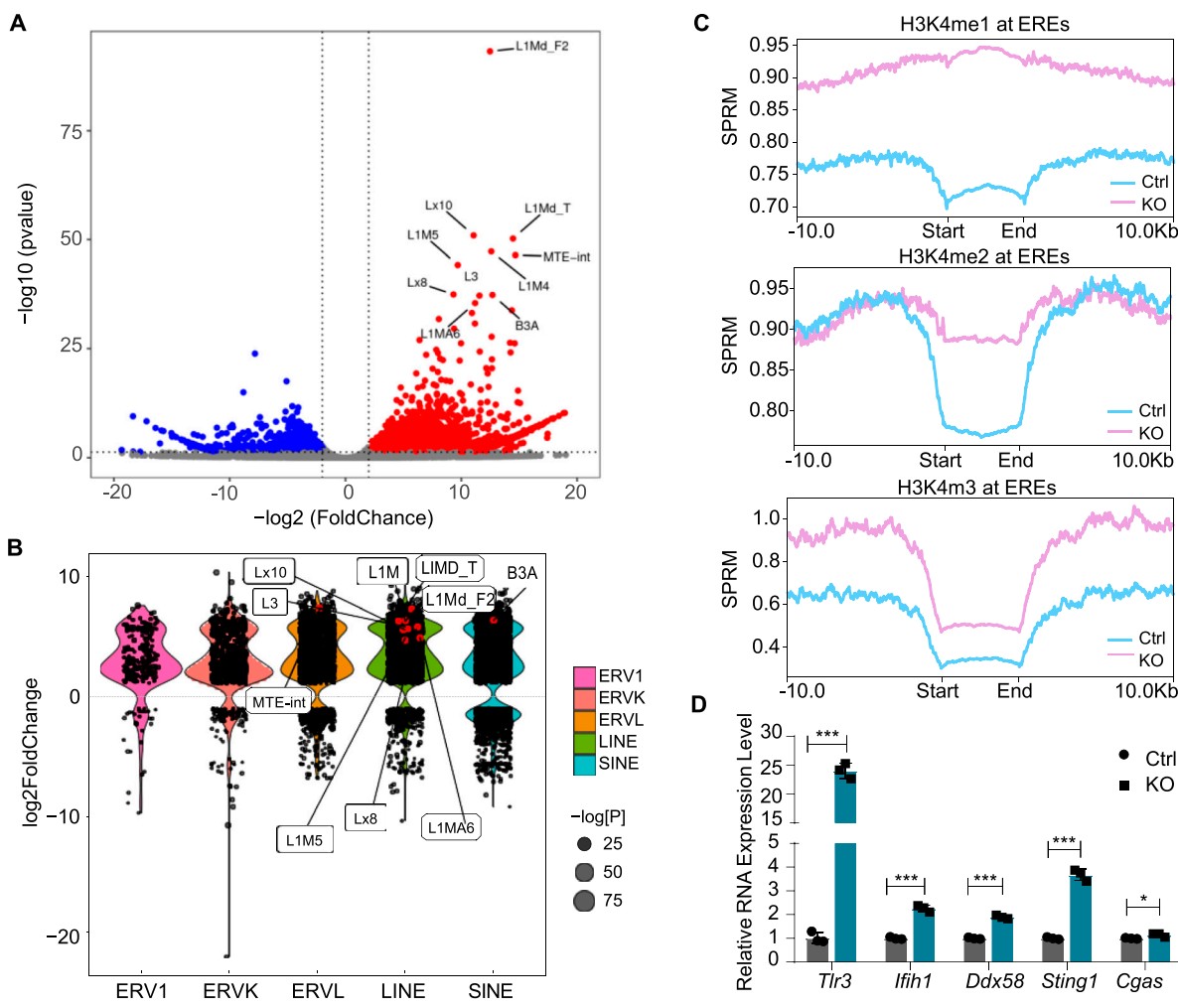

**Figure 6. Deletion of Lsd1 derepresses EREs.**
**(A)** A volcano plot showing differentially expressed EREs (both forward and reverse strands) in total thymocytes from control and KO mice. Increased loci are shown in red and decreased loci are shown in blue. The top 10 significantly expressed loci are labeled. **(B)** The differential expression of ERE classes comparing thymocytes from control and KO mice. **(C)** H3K4me1, H3K4me2, and H3K4me3 ChIP-seq signals at genomic loci of the up-regulated EREs in thymocytes from KO mice compared with those from control mice. **(D)** The expression of the RNA and DNA sensors in control and KO thymocytes were analyzed by qPCR. Control: *Lsd1^{wt/fl}Lck-Cre*, KO: *Lsd1^{fl/fl}Lck-Cre*. The error bars represent the SD between triplicates in one of three experiments. *P < 0.05, **P < 0.01, ***P < 0.001, as determined by unpaired *t* test.

We next investigated the biological effects of the Lsd1 deletion-induced viral mimicry state and IFNγ signaling activation. We observed that compared with those in control mice, the percentages of TCR^{hi}CD69^+CD24^+ immature T cells in KO mice were decreased, whereas those of TCR^{hi}CD69^−CD24^− mature T cells were increased, indicating a premature state of thymic T cells (Fig S6). Moreover, in addition to the innate memory phenotype of CD8 thymocytes mentioned before, we found elevated innate memory scores in all Lsd1-deleted T subgroups by analyzing the scRNA transcriptome (Fig S4B). In addition, the mature T cells in the periphery of KO mice also showed an effector/memory phenotype in the absence of antigen stimulation (Fig S4C). It has been indicated that up-regulation of CD8 expression is associated with the innate memory phenotype in thymic CD8 SP cells. We believe that continuous viral mimicry stimulation and activated IFNγ signaling could be another way to contribute to this phenotype (31).

## TCR formation and selection in Lsd1-deleted thymocytes

TCR signaling is essential for T-cell activation. Notably, we observed decreased expression of TCR receptors on Lsd1-deleted SP cells (Fig S6), suggesting that dysplasia of T-cell function resulted in a premature activation state. Moreover, it is known that TCR recombination events control T-cell development as major checkpoints. Thus, we also investigated the kinetics of TCR recombination by single-cell TCR sequencing. TCRβ transcripts were noticed as early as in the DN3b-ISP subgroup, and TCRα transcripts appeared in large numbers at the CD69^+ DP stage in both control and KO thymi (Fig 7A), indicating that Lsd1 deletion did not disturb the timeline of TCR V(D)J gene recombination events. Furthermore, we analyzed the patterns of the TCR repertoire in different cell types associated with our annotation.

For TCRβ, we observed recombination of the D1 gene with J1 and J2 segments and the D2 gene with J2 segments, whereas nearly no

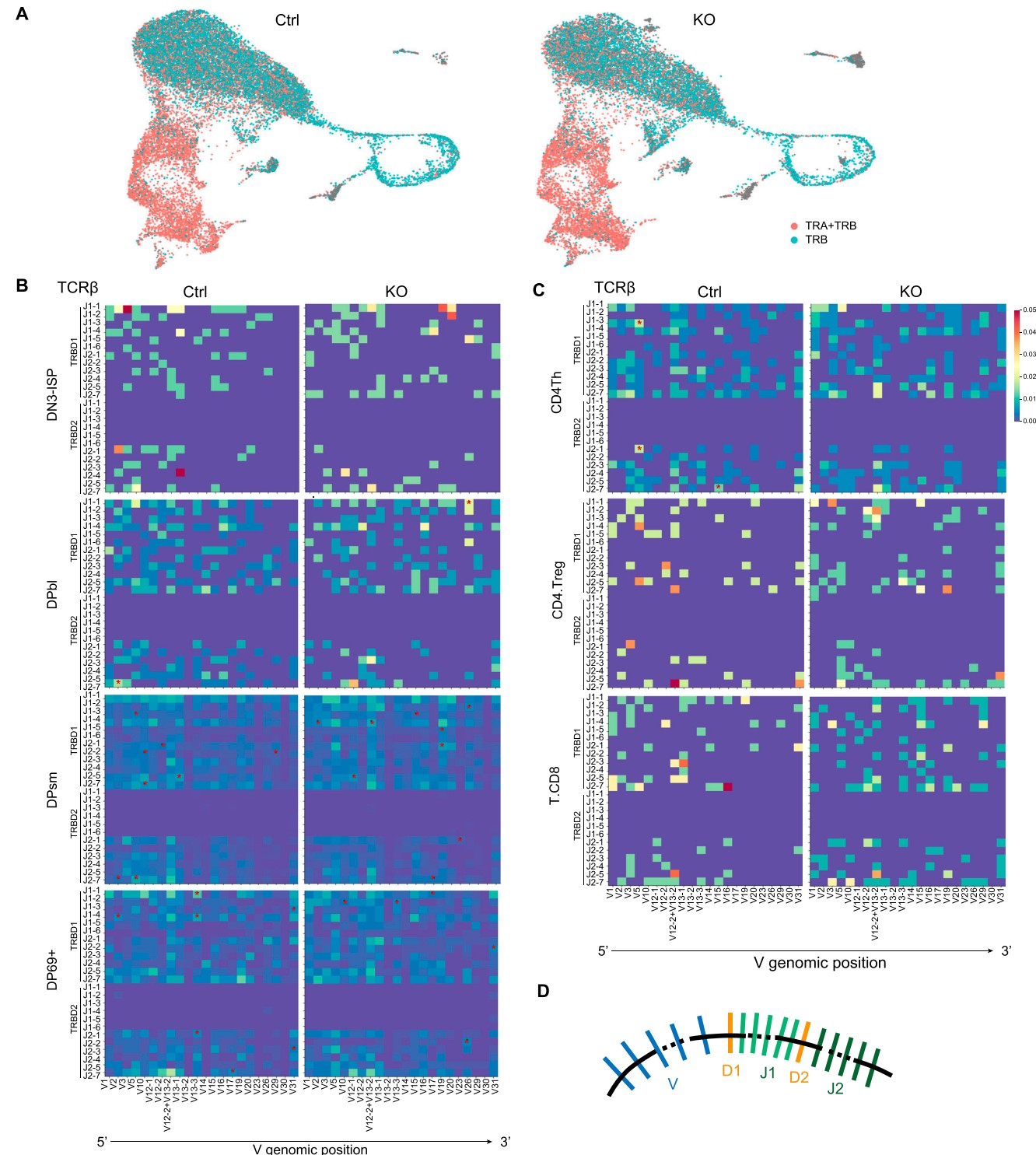

**Figure 7. TCRβ repertoire revealed by scRNA-seq.**
**(A)** UMAP visualization of scRNA-seq data derived from control and KO mice. Cells were color coded by TCR receptors; each dot represents one cell. TRA, TCR α chain; TRB, TCR β chain. **(B, C)** Heatmap showing the proportion of each TCRβ V-D-J pattern present at progressive stages during T-cell development. The red asterisk indicates significantly higher usage compared with the other group. **(D)** Schematics illustrating the genomic location of the Vβ, D, and Jβ gene segments.

use of D2-J1 (Fig 7B and C) was observed in either control or KO thymocytes. This should be associated with their position in the genome, where D2 is located after J1 (32) (Fig 7D). The Vβ-D-Jβ

diversity increased after the development process in both control and KO thymocytes (Fig 7B). For TCRα, we found that the proximal V segments and J segments recombined first during development;

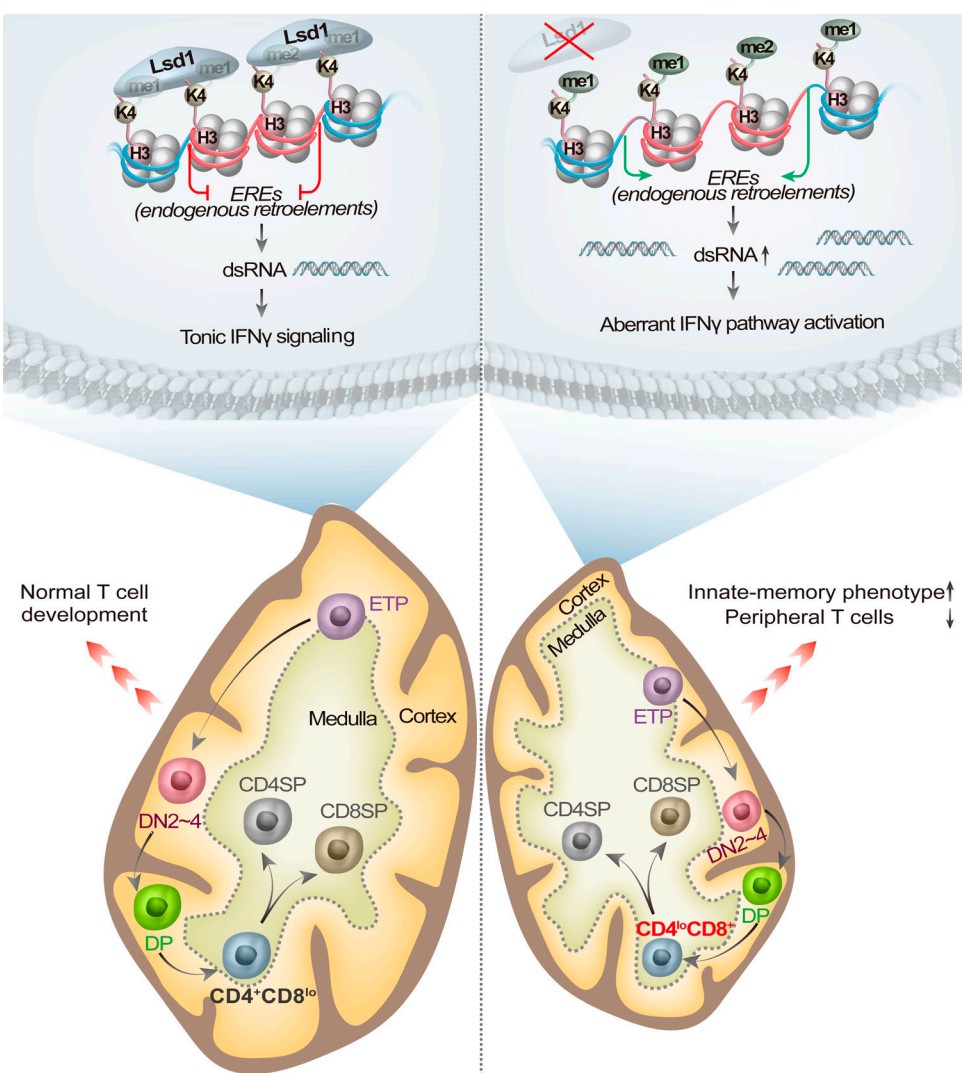

**Figure 8. Proposed model for the role of Lsd1 in thymocyte development.**
Under normal conditions (left), Lsd1 demethylates H3K4me1 and H3K4me2 at the gene loci of a group of EREs and represses their transcription. The thymocytes maintain tonic IFNγ signaling for normal maturation. After the deletion of Lsd1 in thymocytes (right), H3K4me1 and H3K4me2 modifications at the gene loci of a group of EREs are increased, which leads to an accumulation of dsRNAs and aberrantly activates IFNγ signaling at all developmental stages of thymocytes. In the tissue level, deletion of Lsd1 causes severe thymic atrophy with cortex atrophy and medulla expansion. Decreased numbers of peripheral T cells were also observed. Notably, the programmed down-regulation of CD8 expression at the DP→CD4⁺CD8$^{lo}$ stage is disrupted, and an innate memory phenotype is induced in both thymic and peripheral T cells.

this was followed by the recombination of the distal segments (Fig S7), as previously described in humans (25). For both TCRβ and TCRα, contractions of their repertoire during the transition from DP to SP were observed. In addition, different V(D)J patterns were shown in the developed CD4.Th, Treg, and CD8 cells (Figs 7C and S8). These changes suggest differences between Vβ and Vα genes' affinity to MHC molecules and self-antigen peptides during positive and negative selection. It has been shown that the loss of Lsd1 causes the preactivation of T cells by derepressing ERE expression. However, generally, we observed similar diversity and TCR repertoires in different cell types of control and KO mice. Several V-D-J patterns were found to be used differently in DP thymocytes. However, the differences were diminished in SP thymocytes after the "selection" process. That is, unlike the conventional clonal expansion of TCRs after foreign antigen stimulation, endogenous ERE stimulation did not cause a bias in the selection of TCR clones.

Together, the findings indicated that conditional deletion of Lsd1 in immature T cells led to severe thymic atrophy and disrupted the numbers and proliferation of mature T cells in the periphery. The programmed down-regulation of CD8 at the DP→CD4⁺CD8$^{lo}$ stage was disturbed. Moreover, a subgroup of EREs was derepressed with increased H3K4me1/2 modification, inducing the activation of IFNγ signaling at all developmental stages. An innate-memory phenotype of thymic and peripheral T cells was promoted, but no significant shift in the TCR repertoire was found (Fig 8).

## Discussion

In this study, we found that Lsd1 is critical for normal T-cell development and the maintenance of the peripheral T-cell pool. The cellularity of the KO thymus decreased by half; thus, significant thymic atrophy was observed, consistent with that observed in *Lsd1*$^{fl/fl}$*Cd2-Cre* mice (21). DN and DP cells are present in the cortex. Although their proportions were not disturbed, their decreased

**Life Science Alliance**

absolute numbers led to a remarkably atrophic cortex in the KO thymus; and the increased thymic B and DC cells, which are positioned in the cortico-medullary junction and medullary region, respectively, may contribute to the medullary expansion. Furthermore, scRNA-seq revealed similar frequencies of different T-cell subgroups in control and KO mice, except Tregs and NKTs. Interestingly, CD8 expression could not be down-regulated in a timely manner at the intermediate SP stage (CD4+CD8loCCR9+) in KO mice, whereas CD4 expression unexpectedly decreased. However, it did not redirect the following development path to CD4 or CD8 SP cells, as similar CD4/CD8 ratios were observed in the thymus and periphery. Instead, an innate-memory phenotype of CD8+ cells was found. This is consistent with a previous study in which CD8 was overexpressed in thymocytes (26). However, more investigation into how Lsd1 regulates CD8 and CD4 expression at the DP to CD4+CD8lo stage is needed.

After the loss of Lsd1, the results of DEG and pathway enrichment analyses focused on aberrant activation of IFNγ signaling across all the T-cell subgroups, suggesting that this pathway was closely regulated by Lsd1 during T-cell development. Consistently, overexpression of IFN response genes has been observed after the deletion of Lsd1 via Cd2-Cre (22). However, in that study, it remained unclear why the overexpressed genes in Lsd1-deleted thymocytes were not always associated with increased H3K4 methylation, and the authors concluded that Lsd1 indirectly affects their expression. Here, our study further revealed a subgroup of EREs that exhibited increased H3K4me1/2 modification in the KO thymus and could be a direct target of Lsd1. Unlike other viruses, EREs are ancient retroviruses that integrate into host genomic DNA in a germline cell and are inherited by the host's offspring. The epigenetic regulation of ERE transcription in mammalian germ cells and early embryonic development is well documented (33, 34). Recent investigations have also demonstrated that the regulation of ERE transcription plays an important role in human and mouse cancer cells (16, 35, 36). However, little is known about its function in the development of immune cells. Pathogen-induced thymus atrophy is a common phenomenon in infectious diseases, featuring contracted thymic parenchyma and reduced cell numbers (37). Here, we reported that under specific pathogen-free conditions, the loss of Lsd1 in thymocytes caused a similar phenotype by derepressing the expression of a subgroup of EREs. The viral mimicry state induced a general IFN response in thymocytes and increased IFN-γ expression levels. Previous investigations have shown that up-regulation of IFN signaling and increased IFN-γ secretion in the thymic microenvironment are involved in the mechanisms underlying infection-induced thymus atrophy (37). As increased apoptosis has been observed in Lsd1-deleted thymocytes, there could be IFN-γ induced cell death causing thymic atrophy in the KO thymus. On the other hand, T-cell development requires tonic type I IFN signaling. IFN-β is constitutively expressed in thymic medullary epithelial cells, and maturing medullary thymocytes respond to constitutively produced IFN targeting STAT1 and IRF7 (38, 39, 40, 41). Overall, maintaining IFN signaling at a moderate level is necessary for the normal development and maturation of thymic T cells. Thus, Lsd1 acts as an important controller of ERE-IFN signaling.

TCR signaling plays a critical role in programmed T-cell development and is highly associated with T-cell activation. Combined with the scRNA-seq data, we were able to analyze the TCR repertoire among different cell types in the mouse thymus. The initial recombination showed a strong bias, whereas the diversity increased along with DP blasting. Selection is known to shape the TCR repertoire, and enrichment and deletions of TCR V(D)J patterns could be observed at the SP stages. However, the pre-activation state after Lsd1 deletion neither disturbed the timeline of TCR rearrangement nor reshaped the TCR repertoire of SP cells, which may have been because of the innate response to ERE expression rather than acquired immunity against a specific foreign antigen.

Peripheral T cell numbers were markedly reduced after the deletion of Lsd1, consistent with the findings of a previous study on *Lsd1fl/flCd2-Cre* mice (21). They reported a defect in mature SP cell emigration with reduced expression of the emigration marker S1pr1 on mature SP cells. Moreover, we found reduced CD3 expression and impaired proliferation capacity of the peripheral T cells in KO mice, which may have also contributed to the atrophic peripheral T-cell pool. It would be interesting to gain a deeper understanding of the impact of Lsd1 on the function of mature T cells. Recently, a research group knocked out Lsd1 by crossing *Cd4-Cre* transgenic mice with *Lsd1*-floxed mice. Because Lsd1 significantly decreased beginning at the mature peripheral T cells in their study, little effect on thymocyte development was found. They demonstrated that Lsd1 loss in peripheral CD8+ T cells resulted in an increased pool of progenitor-exhausted CD8+ T cells, providing a sustained source for more differentiated T cells with a stronger tumor-killing capacity (42). Similarly, a "memory CD8+ T-cell signature" and an enrichment of "positive regulation of IFN-γ production" were observed in these Lsd1-deficient CD8+ tumor-infiltrating cells. Together, these data suggest that Lsd1 is involved in similar biological pathways at different developmental stages of T cells. For CD4+ T cells, we found no change in the percentage of Th cells but an increase in the percentage of Tregs in the thymus. The effect of Lsd1 on the function and differentiation of peripheral CD4+ T cells is still unknown.

Lsd1 is up-regulated in many cancers and plays a key role in carcinogenesis. Numerous Lsd1 inhibitors are undergoing clinical trials for cancer therapy (43). Recently, it was shown that targeting Lsd1 in B16 tumor cells can also regulate host antitumor T-cell immunity. Lsd1 inhibition reinforces dsRNA stress and IFN responses, which up-regulates the expression of MHC-I molecules and the checkpoint receptor PD-L1 in tumor cells (16). Similar pathways have been found in tumor cells controlled by other histone modulators, such as Kdm5b (35). Here, our study reveals that Lsd1 plays an important role in governing the IFN pathway by controlling ERE transcription for the normal development of thymic T cells and sustaining the peripheral T-cell pool, which should be taken into consideration in Lsd1 inhibitor-based cancer therapy. Our findings highlight the impact of the viral mimicry state on the early development of T cells and suggest that early onset infections invading the thymus could have potential adverse effects on T-cell development and the building of the immune system.

# Materials and Methods

## Mice

$Lsd1^{fl/fl}$ mice (loxP-flanked $Lsd1$ allele at exons 5 and 6) (17), $Lck$-$Cre$ mice (44), and YFP-transgenic mice were kind gifts from Dana-Farber Cancer Center, Boston, MA USA. The deletion was reported to be >50% in immature $CD44^+$ DN cells and almost 100% completed in $CD44^-$ DN or DP and SP cells (44). 6–10-wk-old mice were used for the experiments. All mice were bred under specific pathogen-free conditions. All experiments were approved by the Capital Medical University Animal Care and Use Committee.

## Flow cytometry

Single-cell suspensions were prepared from the thymus, spleen, and lymph nodes. Fluorochrome-conjugated antibodies against CD4 (RM4-5; eBioscience), CD8 (53-6.7; BD Biosciences), CD3 (17A2; BioLegend), CD44 (IM7; BD Biosciences), CD62L (MEL-14; BioLegend), B220 (RA3-6B2; BioLegend), NK1.1 (PK136; BD Biosciences), CD122 (TM-$\beta$1; BD Biosciences), CD69 (H1.2F3; eBioscience), TCR$\beta$ (H57-597; eBioscience), and CD24 (M1/69; eBioscience) were used for staining. FVS510 (BD Biosciences) was used for dead/live cell staining. For intracellular staining, cells were first stained with surface antibodies and then fixed and permeabilized with freshly prepared fixation/permeabilization working solution (BD Biosciences) according to the manufacturer's instructions. Then, the cells were stained with anti-IFN$\gamma$ (XMG1.2; eBioscience), anti-H3K4me1 (D1A9; CST) or anti-H3K4me2 (Y47; Abcam) diluted in permeabilization buffer. Data were acquired on an LSRFortessa (BD Biosciences) and analyzed with FlowJo Software (version 10.6.2). For cell sorting, single-cell suspensions isolated from the thymus were sorted on a FACSAria II (BD Biosciences).

## Histological analysis

Thymi harvested from control and $Lsd1^{fl/fl}Lck$-$Cre$ mice were fixed in 10% formalin solution and embedded in paraffin. Haematoxylin–eosin (HE) standard staining was performed by Servicebio Technology Co., Ltd.

## RNA extraction and qPCR

Whole thymi were dissected, and total cellular RNA was extracted by RNA Isolater Total RNA Extraction Reagent (Vazyme) following the manufacturer's instructions. Total RNA (1 $\mu$g) was reverse transcribed using HiScript QRT SuperMix (+gDNA Wiper) (Vazyme) to generate cDNA. With ChamQ SYBR qPCR Master Mix (Vazyme), qPCR was performed on the Real-Time PCR detection system. All data were normalized to $\beta$-actin mRNA levels, and the $2^{-\triangle CT}$ method was used to calculate the expression levels of target mRNAs. The primers used in the qPCR analysis are presented in Table S1.

## T-cell proliferation

T-cell proliferation was assessed by CFSE dilution with FACS (45, 46). Briefly, a single-cell suspension of splenocytes was prepared and labeled with CFSE (CellTrace CFSE Cell Proliferation Kit; Thermo Fisher Scientific). The labeled cells were cultured in 96-well plates precoated with 5 $\mu$g/ml anti-CD3 monoclonal antibodies and 5 $\mu$g/ml anti-CD28 monoclonal antibodies. After 72 h, splenocytes were stained with APC-Cy7 anti-mouse CD4 (GK1.5), Percp-Cy5.5 anti-mouse CD8 (53-6.7), and FVS510 to exclude dead cells. Non-CFSE-labeled splenocytes and unstimulated splenocytes were used as controls.

## Bulk RNA-seq data processing

The raw reads were aligned to the reference genome mm10 using STAR aligner (version 2.7.9a) with the parameters "--clip3pNbases=0 −clip5pNbases=10 10 −winAnchorMultimapNmax 200 −outFilterMultimapNmax 100 −outSAMstrandField intronMotif". The repeatmasker annotation GTF file for mm10 was obtained using the UCSC table browser (47). NCBI RefSeq mm10 was used for gene annotation. Reads falling in either annotated repeat regions or genes were counted using featureCounts (48) (version 2.0.1) with the parameters "-p -B -M -t exon -s 2" and differential accessibility analysis was performed with DESeq2 (49) (version 1.34.0) with default parameters. Repeat regions with the same "gene_id" but different loci were treated as different repeat regions. Repeat regions with $P$-values < 0.05 and $|\log_2(FC)|$ values > 0.5 were considered differentially expressed. In addition, the $\log_2(FC)$ values were calculated by subtracting the $\log_2$-transformed mean counts in each group.

## scRNA-seq data processing

The raw sequencing data of the thymus cells from control and Lsd1 KO mice were processed using Cell Ranger software (version 6.0.1) against the GRCm38 mouse reference genome with the default parameters. First, we filtered low-quality cells with detected gene numbers between 200 and 4,000 and less than 10% mitochondrial unique molecular identifiers using Seurat (version 4.0.6). Subsequently, we used Scrublet (50) (version 0.2.3) to eliminate doublets among control and KO mice. We used the default parameters for Scrublet (i.e., Eq. min_gene_variability_pctl = 85, n_prin_comps = 30, threshold = 0.25) and detected 12 doublets in the WT mice and three doublets in the KO mice. After removing the doublets, we normalized the gene counts for each cell using the $NormalizeData$ function of Seurat (51) with the default parameters. The top 2,000 highly variable genes were used for principal component analysis.

For downstream data processing, we used the $SelectIntegrationFeatures$ function in Seurat to select features for integration and used the top 2,000 features to identify the anchor cells in control mice and KO mice using the $FindIntegrationAnchors$ function in Seurat. We then used the $IntegrateData$ function in Seurat to integrate the cells from control mice and KO mice. We clustered all the cells based on the integrated gene expression matrix using Seurat with a parameter resolution=1.5 and generated 28 clusters. To display the cells in a two-dimensional space, we

performed PCA on the integrated dataset and used the first 15 principal components (PCs) for UMAP analysis.

### Differential expression analysis

To identify DEGs between two groups of clusters, we used the Wilcoxon rank-sum test in Seurat to evaluate the significance of each gene. $\log_2(FC)$ was calculated by subtracting $\log_2$-transformed mean counts in each group. Genes with a $P$-value < 0.01 and $|\log_2(FC)| > 0.25$ were considered differentially expressed.

### Gene functional annotation

Gene Ontology, gene set enrichment analysis, and KEGG pathway analyses for DEGs were performed using the Metascape ([52]) webtool (www.metascape.org), which supports statistical analysis and visualization of functional profiles for genes and gene clusters.

### Calculation of gene set scores

The gene sets of the innate memory score and ISG score were obtained from the original article ([21], [53]) (Table S2). The gene set scores were calculated with the built-in function *scanpy.tl.score genes* in Scanpy.

### scTCR-seq data processing

The TCR sequence data from 10× Genomics were processed using Cell Ranger software (version 6.0.1) with the manufacturer-supplied mouse VDJ reference genome. For each sample, the output file filtered_contig_annotations.csv, containing TCR $\alpha$- and $\beta$-chain CDR3 nucleotide sequences, was used for downstream analysis. Only those assembled chains that were productive, highly confident, full length, with a valid cell barcode and an unambiguous chain type (for example, alpha) assignment were retained. If a cell had two or more qualified chains of the same type, only the chain with the highest unique molecular identifier count was qualified and retained.

### ChIP-seq data processing

The ChIP-seq data were obtained from a public study in which Lsd1 was knocked out by *Cd2-Cre* recombinase ([21]). Consistent with our study, a reduction in Lsd1 was observed at the DN stage. The raw reads were aligned to the reference genome mm10 using Bowtie2 ([54]) aligner (version 2.2.5). The resultant SAM files were converted to BAM files with samtools (version 1.3.1). Duplicate reads were filtered using Picard. MACS3 ([55]) (version 3.0.0a7) was used to call peaks on the BAM files. The bedGraph files containing signal per million reads produced from MACS3 were converted to bigWig files with the UCSC-toolkit. ChIP-seq signals of ISGs and differentially expressed EREs were extracted and visualized with the deepTools ([56]) (version 3.5.1) command *computeMatrix* and *plotProfile* from bigWig files.

### Statistical analyses

Statistical analyses were performed using GraphPad Prism software (version 8.0). The statistical significance was determined with $t$ test. A $P$-value of less than 0.05 was considered statistically significant. For scRNA-seq data, statistical analysis was performed with the Python (version 3.8.10) package "scipy" (version 1.7.0).

## Data Availability

The accession numbers for the raw data of scRNA-seq and strand-specific total RNA-seq are GSA:CRA007488 and CRA007498, respectively.

## Supplementary Information

## Acknowledgements

We thank all the faculties at the Flow Cytometry Core of Capital Medical University for assistance with flow cytometry sorting and analyzing. We thank Han Yan (Tianjin Medical University) for helping with the model figure drawing. We thank the bioinformatics support from Genewiz Company. We thank the USTC supercomputing center and the School of Life Science Bioinformatics Center for providing computational resources for this project. This work was supported by the National Natural Science Foundation of China (grant# 32270635, 81972652 and 81171899 to X Wang, grant# 82201918 to M Xia, grant, #32022014 to P Chen, grant#91940306, T2125012, 31970858, and 31771428 to K Qu), the Ministry of Science and Technology of People's Republic of China (grant# 2014CB910100 to X Wang), Scientific Research Common Program of Beijing Municipal Commission of Education (grant#KM201910025026 to M Xia), CAS Project for Young Scientists in Basic Research (grant# YSBR-005 to K Qu), and Fundamental Research Funds for the Central Universities (grant# YD2070002019, WK9110000141, and WK2070000158 to K Qu).

### Author Contributions

M Xia: conceptualization, data curation, formal analysis, funding acquisition, investigation, visualization, and writing—original draft.
B Wang: conceptualization, data curation, formal analysis, and visualization.
W Sun: software, formal analysis, and visualization.
D Ji: validation.
H Zhou: investigation.
X Huang: conceptualization and validation.
M Yu: conceptualization and resources.
Z Su: conceptualization.
P Chen: conceptualization, supervision, funding acquisition, methodology, project administration, and writing—review and editing.
K Qu: conceptualization, data curation, software, supervision, funding acquisition, methodology, and writing—review and editing.

X Wang: conceptualization, resources, supervision, funding acquisition, project administration, and writing—review and editing.

## Conflict of Interest Statement

The authors declare that they have no conflict of interest.

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
