## [Reviewer comments · Life Science Alliance]

Life Science Alliance

Lsd1 safeguards T-cell development via suppressing endogenous retroelements and interferon responses

Miaoran Xia, Bingbing Wang, Wujianan Sun, Dengyu Ji, Hang Zhou, Xuefeng Huang, Minghang Yu, Ziyang Su, Ping Chen, Kun Qu, and Xi Wang

DOI: <https://doi.org/10.26508/lsa.202302042>

Corresponding author(s): Xi Wang, Capital Medical University and Kun Qu, University of Science and Technology of China

Review Timeline:

Submission Date:	2023-03-15
Editorial Decision:	2023-04-28
Revision Received:	2023-06-19
Editorial Decision:	2023-06-22
Revision Received:	2023-06-27
Accepted:	2023-06-29

Transaction Report:

April 28, 2023

Re: Life Science Alliance manuscript #LSA-2023-02042-T

Prof. Xi Wang
Capital Medical University
10 Xitoutiao, Youanmenwai, Fengtai District
Beijing 100069
China

Dear Dr. Wang,

Thank you for submitting your manuscript entitled "Lsd1 safeguards T-cell development via suppressing endogenous retroelements and interferon responses" to Life Science Alliance. The manuscript was assessed by expert reviewers, whose comments are appended to this letter. We invite you to submit a revised manuscript addressing the Reviewer comments.

Thank you for this interesting contribution to Life Science Alliance. We are looking forward to receiving your revised manuscript.

Sincerely,

B. MANUSCRIPT ORGANIZATION AND FORMATTING:

Reviewer #1 (Comments to the Authors (Required)):

In this study, Xia et al describes a critical role of Lsd1 in T cell development. Deletion of Lsd1 in thymocytes causes thymic atrophy and reduced peripheral T cell proliferation. Bulk and single cell RNA-seq reveal aberrant depression of EREs and consequently activation of the interferon pathway. Furthermore, loss of Lsd1 also induces an innate memory phenotype in both thymic and peripheral T cells without affecting TCR rearrangement. Overall, this is a well-executed study that identifies Lsd1 as an important regulator of ERE homeostasis in early T cell development. The methods are sound, results are clearly described, and conclusions are well supported by experimental evidence. The paper is suitable for publications with a few clarifications.

1. Can the authors perform Western blots of H3K4me1 and H3K4me3 on nuclear extracts to show their global changes upon Lsd1 KO?
2. Fig. 5E and 6D: are the gene expression changes caused by H3K4me1/3 alterations on promoters?
3. Also, it would be interesting to see if changes in H3K4 methylation lead to changes on H3K9 methylation or histone acetylation on these genes.

Reviewer #2 (Comments to the Authors (Required)):

Xia et al. studied the role of LSD1 in thymocyte development, by deleting Lsd1 at the DN stage in mice. They found that the mice exhibited thymic atrophy, impaired T cell proliferation, and impaired CD8 downregulation. Authors identified aberrant expression of EREs and subsequent IFN response signaling as possible causes of altered thymocyte differentiation under Lsd1 depletion. Authors concluded that LSD1 plays an important role in early T-cell development.

Overall, the study is well-designed and the experimental data are clearly presented. The findings would contribute to understanding how thymocyte development is epigenetically regulated, and how the expression of EREs influence the course of development. There are several concerns that need to be addressed.

1. Lsd1-depletion in DN cells led to atrophy (Fig. 1A) and abnormal architecture (Fig. 1D) in the thymus. These tissue-level changes cannot be explained by the observed characteristics of Lsd1-KO cells (impaired T cell proliferation and development). Discussion on this matter is needed.
2. The authors concluded that LSD1 does not directly regulate IFN-stimulated genes (ISGs), because the levels of H3K4me1 and me2 were lowered in Lsd1-KO cells (Fig. 5F). However, because there was a clear increase in H3K4me3 levels at both TSS and enhancers, it is possible that LSD1 regulates these genes via the maintenance of H3K4 methylation status. In particular, H3K4me3 was found at overactivated enhancers, when the maintenance of H3K4me1 was lost (DOI: 10.1016/j.cell.2016.02.064). To further assess the possible involvement of LSD1 in ISGs regulation, authors should analyze H3K4 methylation status at individual gene locus (e.g., genes shown in Fig. 5E).
3. It is difficult to evaluate the genotype effects on TCR repertoire in Fig. 7. Any quantitative analysis possible?

Reviewer #3 (Comments to the Authors (Required)):

"Lsd1 safeguards T-cell development via suppressing endogenous retroelements and interferon responses" by Xia et al. The study is well written with appropriate recognition of prior work, appropriate techniques have been used and the presentation of data is excellent. The results are clearly described, however this reviewer has concerns about the interpretation of the data presented, how the results gel together and the advance provided by the study as a whole.

Figures 1, 2, 3 and 4 are largely consistent with previously published work using Lsd1^{fl/fl} CD2-Cre mice. The authors could clarify the distinctions.

Figure 5; the authors state "we can conclude that the DEG and their related pathways are associated with the activation of the IFN response".

It is not clear from the data shown if or how the increased expression of IFN γ , but decreased expression of IFN1 and IFN111, affects the expression of the other DE genes.

The fact that IFN1 and IFN111 exhibited decreased expression in KO is not consistent with the statement on page 9 that "consistent with a previous report demonstrating that knockdown of Lsd1 in immature T cells results in the overexpression of gene involved in the IFN/viral response". If you mean IFN γ then please clarify this. This makes the progression into the analyses of H3K4me1/2 confusing (Figure S5).

The use of ERE is unusual, RE is typically used for retroelements that are endogenous.

In the CHIP Seq analyses, the loci of upregulated RE's has increased H3K4me1 and 2 in KO mice. Was that not true for the expressed RE's in "WT" mice?

In addition, these data do not provide direct evidence that Lsd1 targets RE's as stated by the authors.

The authors conclude that Lsa1 derepresses REs, which results in aberrant activation of the IFN response. However, the data do not demonstrate this. It is difficult to determine the relationship between REs and genes involved in the IFN response. There is no evidence provided that REs activate the IFN response. They may be upregulated as a part of a broader locus that contains non-coding and coding components. For instance; because REs are cis-regulatory elements and form part of lnc RNA's one might expect to see the increased expression of REs in addition to protein coding genes.

Manuscript #LSA-2023-02042-T

Response to Reviewers

Reviewers' Comments to the Authors:

Reviewer #1 :

In this study, Xia et al describes a critical role of Lsd1 in T cell development. Deletion of Lsd1 in thymocytes causes thymic atrophy and reduced peripheral T cell proliferation. Bulk and single cell RNA-seq reveal aberrant depression of EREs and consequently activation of the interferon pathway. Furthermore, loss of Lsd1 also induces an innate memory phenotype in both thymic and peripheral T cells without affecting TCR rearrangement. Overall, this is a well-executed study that identifies Lsd1 as an important regulator of ERE homeostasis in early T cell development. The methods are sound, results are clearly described, and conclusions are well supported by experimental evidence. The paper is suitable for publications with a few clarifications.

1. Can the authors perform Western blots of H3K4me1 and H3K4me3 on nuclear extracts to show their global changes upon Lsd1 KO?

Author Response: As suggested by the reviewer, we have performed western blots of H3K4me1/2/3 on nuclear extracts to show their global changes upon the deletion of Lsd1. Consistent with our previous data, in which we detected the H3K4me1/2 by flow cytometry (Figure S1D), we observed no difference of H3K4me1 and increased H3K4me2 in KO thymocytes. And there was no change of the H3K4me3 modification as below (Figure A1-1). We have also included the results in revised Figure S1C. Thanks!

Figure A1-1. The H3K4me1/2/3 modification in Ctrl and KO thymocytes.

2. Fig. 5E and 6D: are the gene expression changes caused by H3K4me1/3 alterations on promoters?

Author Response: As pointed out by the reviewer, we have analyzed the H3K4me1/2/3 status at individual locus for the genes shown in Fig. 5E and 6D. We summarized the results below (Table A1-1). The tracks of individual genes are also attached (Figure A1-2).

As shown, there was a global increase in the H3K4me3 modification of these genes in KO thymocytes, but the H3K4me1/2 modifications were mainly decreased or unchanged. We thus conclude that the activation of these genes was associated with the H3K4me3 alteration. However, as *Lsd1* demethylates only H3K4me2/1 but not H3K4me3, expression of these genes may not be directly regulated by *Lsd1*.

Table A1-1. The H3K4me1/2/3 alterations in KO thymocytes at individual gene loci

Gene	H3K4me1	H3K4me2	H3K4me3
H2-K1	↓↓	↓↓	↑↑
H2-D1	↓↓	↓↓	↑↑
Irf7	↓↓	↓	↑↑
Irf9	↓↓	No significant change	↑↑
Oas3	No significant change	No significant change	↑
Ifit1	No significant change	↑	↑↑
Stat1	↓	↓	↓
Nfkb1	↑	↓	↑
Tlr3	No significant change	No significant change	↑
Ifih1	No significant change	No significant change	↑
Ddx58	↓	↓	↑
Sting1	↓	No significant change	↑↑
Cgas	↓↓	No significant change	↑↑

Figure A1-2. The H3K4me1/2/3 status at individual gene loci.

3. Also, it would be interesting to see if changes in H3K4 methylation lead to changes on H3K9 methylation or histone acetylation on these genes.

Author Response: Thank you for this great suggestion! We also analyzed H3K9me1/2/3 and H3K27ac on nuclear extracts to show their global changes upon deletion of Lsd1. The results indicated that there were no changes in global H3K9me1/2/3 and H3K27ac status (Figure A1-3). We will perform ChIP analysis to further confirm this on individual genes in our future study. Please find the data below. We have also included the results in revised Figure S1C. Thanks!

Figure A1-3. The H3K9me1/2/3 and H3K27ac modification in Ctrl and KO thymocytes.

Reviewer #2 :

Xia et al. studied the role of LSD1 in thymocyte development, by deleting Lsd1 at the DN stage in mice. They found that the mice exhibited thymic atrophy, impaired T cell proliferation, and impaired CD8 downregulation. Authors identified aberrant expression of EREs and subsequent IFN response signaling as possible causes of altered thymocyte differentiation under Lsd1 depletion. Authors concluded that LSD1 plays an important role in early T-cell development.

Overall, the study is well-designed and the experimental data are clearly presented. The findings would contribute to understanding how thymocyte development is epigenetically regulated, and how the expression of EREs influence the course of development. There are several concerns that need to be addressed.

1. Lsd1-depletion in DN cells led to atrophy (Fig. 1A) and abnormal architecture

(Fig. 1D) in the thymus. These tissue-level changes cannot be explained by the observed characteristics of *Lsd1*-KO cells (impaired T cell proliferation and development). Discussion on this matter is needed.

Author Response: Thank you for pointing this out.

(1) It has been observed that viruses can induce thymic atrophy. Direct and indirect cell death induced by infection are two common causes of thymic atrophy (Luo *et al.*, *Front Immunol.*, 2021, doi: 10.3389/fimmu.2021.652538). One of the indirect cell death pathways is induced by cytokines. For example, IFN- γ secreted by NK cells and activated thymic innate CD8⁺CD44^{hi} SP thymocytes during influenza A, and also has been reported to result in thymic atrophy (Duan *et al.*, *J Gen Virol*, 2015, doi: 10.1099/jgv.0.000276; Liu *et al.*, *Cell Death Dis*, 2014, doi:10.1038/cddis.2014.323). Our study also revealed significantly increased cell apoptosis in the *Lsd1*-KO thymus (shown below, Figure A2-1, included in Figure S1E) and great decrease of the total cell numbers of thymus (Figure 1C), indicating that the viral mimicry state could also lead to thymic atrophy. Moreover, increased IFN γ response in KO mice may also lead to thymic atrophy. We will investigate this in more depth. Discussion on this has been added on page 13, lines 395-400.

*"Previous investigations have shown that upregulation of IFN signaling and increased IFN- γ secretion in the thymic microenvironment are involved in the mechanisms underlying infection-induced thymus atrophy [36]. As increased apoptosis has been observed in *Lsd1*-deleted thymocytes, there could be IFN- γ induced cell death causing thymic atrophy in the KO thymus."*

Figure A2-1. Apoptosis in the thymus.

(2) The abnormal architecture was featured by a remarkably atrophic cortex and expanded medullary regions. The thymic tissue has darker cortical regions and lighter medullary areas under the microscope. This difference in brightness is attributed to the difference in their cell density (Abbas A K, Lichtman A H, Pillai S. *Cellular and molecular immunology*. Elsevier Health Sciences, 2018; Murphy K, Weaver C. *Janeway's immunobiology*. Garland Science, 2016). The darker cortex consists of immature thymocytes, including DN and DP cells. It has much more thymocytes compared to the medulla. In this

study, the total numbers of thymus were greatly reduced (Figure 1C). Although the proportions of DN and DP cells remained (Figure 1F), the absolute numbers of DN and DP cells located in the cortex decreased significantly (Figure A2-2). This can explain the remarkably atrophic cortex. Thymic B cells are present at the cortico-medullary junction (Fehervari, Nat Immunol, 2013, doi.org/10.1038/ni.2777), and DCs are positioned in the medullary region of the thymus (Oh et al., Immune Netw, 2015, doi: 10.4110/in.2015.15.3.111). In this study, although there was no change in the total number of medullary SP cells (Figure A2-2), we observed significantly increased thymic B cells and DCs after Lsd1 deletion (Figure 3E & S3A), which can explain the medullary expansion. As the reviewer's suggestion, we have added the discussion on Page 12, lines 361-365.

"DN and DP cells are present in the cortex. Although their proportions were not disturbed, their decreased absolute numbers led to a remarkably atrophic cortex in the KO thymus. And the increased thymic B and DC cells, which are positioned in the cortico-medullary junction and medullary region respectively, may contribute to the medullary expansion."

Figure A2-2. Absolute T cell numbers in the cortex and medulla.

2. The authors concluded that LSD1 does not directly regulate IFN-stimulated genes (ISGs), because the levels of H3K4me1 and me2 were lowered in Lsd1-KO cells (Fig. 5F). However, because there was a clear increase in H3K4me3 levels at both TSS and enhancers, it is possible that LSD1 regulates these genes via the maintenance of H3K4 methylation status. In particular, H3K4me3 was found at overactivated enhancers, when the maintenance of H3K4me1 was lost (DOI: 10.1016/j.cell.2016.02.064). To further assess the possible involvement of LSD1 in ISGs regulation, authors should analyze H3K4 methylation status at individual gene locus (e.g., genes shown in Fig. 5E).

Author Response: As suggested by the reviewer, we have analyzed the H3K4me1/2/3 status at individual gene locus for the genes shown in Fig. 5E and

6D. We summarized the results below (Table A2-1). The tracks of individual genes are also attached (Figure A2-3).

As shown, there was a global increase in the H3K4me3 modification of these genes, but the H3K4me1/2 modifications were mainly decreased or unchanged. We can conclude that the activation of these genes was associated with the H3K4me3 alteration. However, as Lsd1 can demethylate only H3K4me2/1 but not H3K4me3, these gene expressions may be indirectly targeted by Lsd1. On the other hand, it is possible that Lsd1 affects their H3K4me3 status by cooperating with other epigenetic enzymes other than its demethylase activity.

Table A2-1. The H3K4me1/2/3 alterations in KO thymocytes at individual gene loci

Gene	H3K4me1	H3K4me2	H3K4me3
H2-K1	↓↓	↓↓	↑↑
H2-D1	↓↓	↓↓	↑↑
Irf7	↓↓	↓	↑↑
Irf9	↓↓	No significant change	↑↑
Oas3	No significant change	No significant change	↑
Ifit1	No significant change	↑	↑↑
Stat1	↓	↓	↓
Nfkb1	↑	↓	↑
Tlr3	No significant change	No significant change	↑
Ifih1	No significant change	No significant change	↑
Ddx58	↓	↓	↑
Sting1	↓	No significant change	↑↑
Cgas	↓↓	No significant change	↑↑

Figure A2-3. The H3K4me1/2/3 status at individual gene loci.

3. It is difficult to evaluate the genotype effects on TCR repertoire in Fig. 7. Any quantitative analysis possible?

Author Response: Yes, we had quantitative analysis on the differential expressed TCR rearrangements, which were marked with red asterisks in Figure 7B, 7C, and Figure S7, S8. We also analyzed the TCR diversity measured by D50 (Figure A2-4) and Shannon entropy (Figure A2-5). No significant difference was observed.

Figure A2-4. TCR diversity measured by D50

Figure A2-5. TCR diversity measured by Shannon entropy

Reviewer #3:

"Lsd1 safeguards T-cell development via suppressing endogenous retroelements and interferon responses" by Xia et al. The study is well written with appropriate recognition of prior work, appropriate techniques have been used and the presentation of data is excellent. The results are clearly described, however this reviewer has concerns about the interpretation of the data presented, how the results gel together and the advance provided by the study as a whole.

1. Figures 1, 2, 3 and 4 are largely consistent with previously published work using Lsd1^{fl/fl} CD2-Cre mice. The authors could clarify the distinctions.

Author Response: Thank you for the comment. We have clarified the distinctions figure by figure as below.

In Figure 1, the result showing the decreased number of thymocytes (Figure 1C) is consistent with the previous work. In addition, we showed the appearance (Figure 1A), weight (Figure 1B), and tissue architecture (Figure 1D) of the thymus. Moreover, we emphasized the alteration of the CD4/CD8 ratio after Lsd1 deletion (Figure 1G), which was further demonstrated in Figure 4.

In Figure 2, we both observed a reduction of the peripheral T cells. Other than this, we found that the proliferation capacities of both peripheral CD4⁺ and

CD8⁺ T cells were impaired after *Lsd1* deletion (Figure 2E & F). In the previous study, the authors attributed the reduction of the peripheral T cells to the reduced *S1pr1* required for thymocyte emigration on mature SP cells. We think that the impaired proliferation capacity could be one of the reasons. It has been discussed in Result 1 (Page 5, Lines 132-140).

In Figure 3, we performed the single-cell RNA-seq of the total cells in the thymus. We identified the cell types with distinct transcriptomic signatures and got a full view of all the developmental stages. In the previous study, the scRNA-seq was performed on the mixture of sorted DN (CD4⁻ CD8⁻), CD8 ISP (CD8⁺CD103⁻CD69⁻CD5^{lo}), and unsignaled DP (CD4⁺CD8⁺CD69^{lo/-}) thymocytes (ratio of ~2:1:1), which was identified by surface markers in flow cytometry.

In Figure 4, we analyzed the alternation of CD4/CD8 expression in intermediate SP cells and identified the surface marker of this group. In the previous study, they didn't study this subgroup.

2. Figure 5; the authors state "we can conclude that the DEG and their related pathways are associated with the activation of the IFN response".

It is not clear from the data shown if or how the increased expression of IFN γ , but decreased expression of IFN I and IFN III, affects the expression of the other DE genes.

Author Response: Thanks for the comment. The DEGs in scRNA-seq were enriched on the IFN response, as shown in Figure 5C. And the interferon-stimulating genes (ISGs) were upregulated significantly, as shown in Figure 5D. From our results, we observe that IFN γ , but not IFN I and III, is the main type of IFN that expressed by the thymocytes (shown below, Figure A3-1), which has also been mentioned in the manuscript on Page 9, "the IFN-I gene (*Ifna1*, *Ifnb*) and IFN-III gene (*Il28b*) were expressed at low levels". This is consistent with the findings that IFN I and III were mainly secreted by epithelial cells, fibroblasts, and innate immune cells (e.g., pDCs, macrophage) (Barrat *et al.*, *Nat. Immunol.*, 2019, doi: 10.1038/s41590-019-0466-2), while T cells mainly secrete IFN γ under stimulus (Ivashkiv, *Nat. Rev. Immunol.*, 2018, doi:10.1038/s41577-018-0029-z). Although distinct cell types produce type I, III, and II IFNs, the genes or signatures controlled by these IFNs overlap substantially (Barrat *et al.*, *Nat. Immunol.*, 2019, doi: 10.1038/s41590-019-0466-2; Liu *et al.*, *PNAS*, 2012, doi.org/10.1073/pnas.1114981109). So we think that IFN γ plays a major role in *Lsd1*-KO thymocytes and affects the expression of the DEGs. We improved our manuscript as shown in page 9, lines 247-252.

*"As known, T cells mainly secrete IFN γ under stimulus [29]. Increased expression of IFN-II (IFN- γ) was determined at both the mRNA and protein levels in KO mice compared to control mice (Figure S5A and Figure S5B), while the IFN-I gene (*Ifna1*, *Ifnb*) and IFN-III gene (*Il28b*) were expressed at much lower levels and even showed some decrease in expression (Figure S5A) in KO thymocytes."*

Figure A3-1. The qPCR analysis of IFN gene expression in sorted thymocytes from Ctrl mice.

3. The fact that IFN α and IFN γ exhibited decreased expression in KO is not consistent with the statement on page 9 that "consistent with a previous report demonstrating that knockdown of Lsd1 in immature T cells results in the overexpression of gene involved in the IFN/viral response". If you mean IFN γ then please clarify this. This makes the progression into the analyses of H3K4me1/2 confusing (Figure S5).

Author Response: Thank you for pointing this out. As mentioned above, IFN α and γ were expressed at low levels in T cells, and IFN γ plays a major role. As suggested, we have clarified this in the revised manuscript by replacing "IFN" with "IFN γ ". As lots of IFN-responsive genes were upregulated, revealed by the scRNA-seq, we then analyzed the active marker H3K4me1/2 on these gene locus as Lsd1 is a well-known H3K4me1/2 demethylase.

4. The use of ERE is unusual, RE is typically used for retroelements that are endogenous.

Response: Yes, RE represents retroelements that are endogenous. And ERE, short for endogenous retroelements (EREs), is also commonly used in the literature (*Nat Immunol*, 2014, doi.org/10.1038/ni.2872; *Nat Rev Immunol*, 2016, doi.org/10.1038/nri.2016.27; *Annual Review of Genetics*, 2019, doi.org/10.1146/annurev-genet-112618-043717). They are referred to as remnants of transposable elements that integrated into host germline DNA millions of years ago. We use 'endogenous' to emphasize that they were the product of the host genomic DNA rather than the exogenous viruses.

5. In the CHIP Seq analyses, the loci of upregulated RE's has increased H3K4me1 and 2 in KO mice. Was that not true for the expressed RE's in "WT" mice?

In addition, these data do not provide direct evidence that Lsd1 targets RE's as stated by the authors.

Author Response: As interested by the reviewer, we analyzed the H3K4me1/2

status of expressed REs in WT mice. We calculated the Z-score of the RE expression levels in the three Ctrl mice. Then we defined those REs with Z-score > 0 as the expressed REs (marked as Ctrl-Up) and those with Z-score < 0 as Ctrl-Down. The results are shown below (Figure A3-2). The H3K4me1/2 modifications are increased significantly in the expressed REs in WT mice.

Figure A3-2. The H3K4me1/2 modifications in the expressed REs in WT mice.

In this study, we found that lots of REs were upregulated in thymocytes after *Lsd1* deletion (Figure 6A & 6B). And the ChIP-seq analysis showed that the loci of upregulated REs had a great increase in H3K4me1 and H3K4me2 levels (Figure 6C). As *Lsd1* is well-known as a demethylase of the active modification H3K4me1/2, we speculated that there is a high probability that *Lsd1* can target REs. Nevertheless, we need more direct evidence, such as *Lsd1* ChIP analysis on RE loci, which we will perform later in our further study.

6. The authors conclude that *Lsd1* derepresses REs, which results in aberrant activation of the IFN response. However, the data do not demonstrate this. It is difficult to determine the relationship between REs and genes involved in the IFN response. There is no evidence provided that REs activate the IFN response. They may be upregulated as a part of a broader locus that contains non-coding and coding components. For instance; because REs are cis-regulatory elements and form part of lnc RNA's one might expect to see the increased expression of REs in addition to protein coding genes.

Author Response: Thank you for pointing this out. Inspired by you, we read the literature and find that some ERVs can act as IFN-inducible enhancers in macrophages (Chuong et al., *Science*, 2016, doi.org/10.1126/science.aad5497) and cis-regulatory elements of Th1 genes (Adoue et al., *Immunity*, 2019, doi.org/10.1016/j.immuni.2019.01.003). However, we still need relevant experimental evidence to support if this is true in early T cells. On the other hand, it has been widely observed that the RE-derived dsRNAs are recognized as invading pathogens, which triggers activation of the interferon signaling and innate immune response. This phenomenon has been termed 'viral mimicry'

(Chiappinelli *et al.*, *Cell*, 2015, doi:10.1016/j.cell.2015.07.011; Buttler CA, Chuong EB. *Immunological reviews*, 2022, doi: 10.1111/imr.13042). The signaling cascades start with a variety of sensors, such as RIG-I, MDA5, cGAS, and TLRs. (Gorbunova *et al.*, *Nature*, 2021, doi.org/10.1038/s41586-021-03542-y) In our study, we also found increased expression of the DNA or dsRNA sensors TLR3, MDA5 (encoded by *Ifih1*), RIG-I (encoded by *Ddx58*), Sting, and C-gas (Figure 6D), suggesting an endogenous response was activated in the KO thymocytes. Recently, lots of publications demonstrated the aberrant activation of IFN response by REs, mainly in cancer cells. Here are some instances:

(1) In 2005, two publications reported at the same time that inhibiting DNA methylation derepressed endogenous elements and caused interferon response in colorectal and ovarian cancer cells, respectively. (Roulois *et al.*, *Cell*, 2015, doi:10.1016/j.cell.2015.07.056; Chiappinelli *et al.*, *Cell*, 2015, doi:10.1016/j.cell.2015.07.011)

(2) Ablation of LSD1 in cancer cells increases ERV expression and decreases expression of RNA-induced silencing complex (RISC) components, which leads to double-stranded RNA (dsRNA) stress and activation of type 1 interferon, stimulating anti-tumor T cell immunity and restrains tumor growth (Sheng *et al.*, *Cell*, 2018, doi.org/10.1016/j.cell.2018.05.052).

(3) KDM5B recruits the H3K9 methyltransferase SETDB1 to repress endogenous retroelements in melanoma. The derepression of these retroelements activates cytosolic RNA-sensing and DNA-sensing pathways and the subsequent type-I interferon response. (Zhang *et al.*, *Nature*, 2021, doi.org/10.1038/s41586-021-03994-2)

As the reviewer's suggestion, more evidence is needed for the relationship between RE expression and the IFN II response. In future studies, we will find a way to silence/overexpress these upregulated REs conditionally in thymocytes, or knock out the dsRNA sensors like Mda5 or Rig-I to block the dsRNA-sensing pathway, to see if there is an alteration of the IFN γ response.

June 22, 2023

RE: Life Science Alliance Manuscript #LSA-2023-02042-TR

Prof. Xi Wang
Capital Medical University
10 Xitoutiao, Youanmenwai, Fengtai District
Beijing, Beijing 100069
China

Dear Dr. Wang,

Thank you for submitting your revised manuscript entitled "Lsd1 safeguards T-cell development via suppressing endogenous retroelements and interferon responses". We would be happy to publish your paper in Life Science Alliance pending final revisions necessary to meet our formatting guidelines.

- please add an ORCID ID for the corresponding secondary author--they should have received instructions on how to do so
- please add the Twitter handle of your host institute/organization as well as your own or/and one of the authors in our system
- you may consider uploading Figure 8 as a Graphical Abstract instead, but this is up to you

A. FINAL FILES:

B. MANUSCRIPT ORGANIZATION AND FORMATTING:

****It is Life Science Alliance policy that if requested, original data images must be made available to the editors. Failure to provide**

original images upon request will result in unavoidable delays in publication. Please ensure that you have access to all original data images prior to final submission.**

The license to publish form must be signed before your manuscript can be sent to production. A link to the electronic license to publish form will be sent to the corresponding author only. Please take a moment to check your funder requirements.

Sincerely,

Reviewer #1 (Comments to the Authors (Required)):

The authors have addressed all my previous concerns and the paper is suitable to publish.

Reviewer #2 (Comments to the Authors (Required)):

There are no further comments from this reviewer.

June 29, 2023

RE: Life Science Alliance Manuscript #LSA-2023-02042-TRR

Prof. Xi Wang
Capital Medical University
10 Xitoutiao, Youanmenwai, Fengtai District
Beijing, Beijing 100069
China

Dear Dr. Wang,

Thank you for submitting your Research Article entitled "Lsd1 safeguards T-cell development via suppressing endogenous retroelements and interferon responses". It is a pleasure to let you know that your manuscript is now accepted for publication in Life Science Alliance. Congratulations on this interesting work.

DISTRIBUTION OF MATERIALS:

Again, congratulations on a very nice paper. I hope you found the review process to be constructive and are pleased with how the manuscript was handled editorially. We look forward to future exciting submissions from your lab.

Sincerely,
